# Annealing Flow Generative Models Towards Sampling High-Dimensional and Multi-Modal Distributions

Dongze Wu [1]   Yao Xie [1]

## Abstract

Sampling from high-dimensional, multi-modal distributions remains a fundamental challenge across domains such as statistical Bayesian inference and physics-based machine learning. In this paper, we propose *Annealing Flow* (AF), a method built on Continuous Normalizing Flow (CNF) for sampling from high-dimensional and multi-modal distributions. AF is trained with a dynamic Optimal Transport (OT) objective incorporating Wasserstein regularization, and guided by annealing procedures, facilitating effective exploration of modes in high-dimensional spaces. Compared to recent NF methods, AF greatly improves training efficiency and stability, with minimal reliance on MC assistance. We demonstrate the superior performance of AF compared to state-of-the-art methods through experiments on various challenging distributions and real-world datasets, particularly in high-dimensional and multi-modal settings. We also highlight AF's potential for sampling the least favorable distributions.

## 1. Introduction

Sampling from high-dimensional and multi-modal distributions is crucial for various fields, including physics-based machine learning such as molecular dynamics (Miao et al., 2015; Salo-Ahen et al., 2020), quantum physics (Carlson et al., 2015; Lynn et al., 2019), and lattice field theory (Jay & Neil, 2021; Lozanovski et al., 2020). With modern datasets, it also plays a key role in Bayesian areas, including Bayesian modeling (Kandasamy et al., 2018; Balandat et al., 2020; Stephan et al., 2017) with applications in areas like computational biology (Stanton et al., 2022; Overstall et al., 2020), and Bayesian Neural Network sampling (Cobb & Jalaian,

2021; Izmailov et al., 2021; Wu et al., 2024).

*MCMC and Neural Network Variants*: Numerous MCMC methods have been developed over the past 50 years, including Metropolis-Hastings (MH) and its variants (Haario et al., 2001; Cornish et al., 2019; Griffin & Walker, 2013; Choi, 2020), Hamiltonian Monte Carlo (HMC) schemes (Girolami & Calderhead, 2011; Bou-Rabee & Sanz-Serna, 2017; Shahbaba et al., 2014; Li et al., 2015; Hoffman et al., 2021; 2014). HMC variants are still considered state-of-the-art methods. However, they require exponentially many steps in the dimension for mixing, even with just two modes (Hackett et al., 2021). More recently, Neural network (NN)-assisted sampling algorithms (Wolniewicz et al., 2024; Bonati et al., 2019; Gu & Sun, 2020; Egorov et al., 2024; Li et al., 2021; Hackett et al., 2021) have been developed to leverage NN expressiveness for improving MCMC, but they still inherit some limitations like slow mixing and imbalanced mode exploration, particularly in high-dimensional spaces.

*Annealing Variants*: Annealing methods (Gelfand et al., 1990; Sorkin, 1991; Van Groenigen & Stein, 1998; Neal, 2001) are widely used to develop MCMC techniques like Parallel Tempering (PT) and its variants (Earl & Deem, 2005; Chandra et al., 2019; Syed et al., 2022). In annealing, sampling gradually shifts from an easy distribution to the target by lowering the temperature. Annealed Importance Sampling (Neal, 2001) and its variants(Zhang et al., 2021; Karagiannis & Andrieu, 2013; Chehab et al., 2024) are developed for estimating normalizing constants with low variance using MCMC samples from intermediate distributions. Recent Normalizing Flow and score-based annealing methods (Arbel et al., 2021; Doucet et al., 2022) optimize intermediate densities for lower-variance estimates, but still rely on MCMC for sampling. However, MCMC struggles with slow mixing, local mode trapping, mode imbalance, and correlated samples issues. These limitations are particularly pronounced in high-dimensional, multi-modal settings (Van Ravenzwaaij et al., 2018; Hackett et al., 2021).

*Particle Optimization Methods*: Recently, particle-based optimization methods have emerged for sampling, including Stein Variational Gradient Descent (SVGD) (Liu & Wang, 2016), and stochastic approaches such as Dai et al. (2016); Nitanda & Suzuki (2017); Maddison et al. (2018);

---

[1]H. Milton Stewart School of Industrial and Systems Engineering (ISyE), Georgia Institute of Technology, USA. Correspondence to: Yao Xie <yao.xie@isye.gatech.edu>.

*Proceedings of the $42^{nd}$ International Conference on Machine Learning*, Vancouver, Canada. PMLR 267, 2025.

Liu (2017); Pulido & van Leeuwen (2019); Li et al. (2023); Detommaso et al. (2018); Maurais & Marzouk (2024). However, many of these methods rely on kernel computations, which scale polynomially with sample size, and are sensitive to hyperparameters.

*Normalizing Flows*: Recently, discrete Normalizing Flows (NFs) (Rezende & Mohamed, 2015) and Stochastic NFs (Wu et al., 2020; Hagemann et al., 2022) have been actively explored for sampling tasks. Discrete NFs sometimes suffer from mode collapse, and methods relying on them (Arbel et al., 2021; Matthews et al., 2022; Gabrié et al., 2021; 2022; Albergo & Vanden-Eijnden, 2023; Brofos et al., 2022; Cabezas et al., 2024; Qiu & Wang, 2024) attempt to mitigate this issue using MCMC corrections or design specialized loss functions. More recently, Fan et al.; Tian et al. (2024) introduced path-guided NFs, which utilize training losses conceptually similar to score matching. However, these methods may require a substantial number of discretized time steps, with score estimation at each step. Besides, the quality of these estimations substantially influences the overall training performance.

Challenges persist with multi-modal distributions in high-dimensional spaces. This paper introduces *Annealing Flow* (AF), a Continuous Normalizing Flow trained with a dynamic Optimal Transport (OT) objective incorporating Wasserstein regularization, and guided by annealing procedures. Our key contributions are as follows:

- We present a dynamic OT objective that greatly reduces the number of intermediate annealing steps and improves training stability, compared to recent annealing-like NFs (Tian et al., 2024; Fan et al.), which require score estimation and matching during training.

- The proposed annealing-assisted procedure is crucial for success on high-dimensional distributions with widely separated modes, outperforming state-of-the-art NF-based methods in challenging experiments with notably fewer annealing steps.

- Theoretically, Theorem 3.3 establishes that the infinitesimal optimal velocity field equals the difference in the score between consecutive annealing densities, a desirable property unique to our dynamic OT objective.

## 2. Preliminaries

*Neural ODE and Continuous Normalizing Flow:* A Neural ODE is a continuous model where the trajectory of data is modeled as the solution of an ordinary differential equation (ODE). Formally, in $\mathbb{R}^d$, given an input $x(0) = x_0$ at time $t = 0$, the transformation to the output $x(1)$ is governed by:

$$\frac{dx(t)}{dt} = v(x(t), t), \quad t \in [0, 1], \tag{1}$$

where $v(x(t), t)$ represents the velocity field, which is of the same dimension as $x(t)$ and is parameterized by a neural network with input $x(t)$ and $t$. The time horizon is rescaled to $[0, 1]$ without loss of generality.

Continuous Normalizing Flow (CNF) is a class of normalizing flows where the transformation of a probability density from a base distribution $\pi_0(x)$ (at $t = 0$) to a target distribution $q(x)$ (at $t = 1$) is governed by a Neural ODE. The marginal density of $x(t)$, denoted as $\rho(x, t)$, evolves according to the continuity equation derived from the ODE in Eq. (1). This continuity equation is written as:

$$\partial_t \rho(x, t) + \nabla \cdot (\rho(x, t) v(x, t)) = 0, \quad \rho(x, 0) = \pi_0(x), \tag{2}$$

where $\nabla \cdot$ denotes the divergence operator.

*Dynamic Optimal Transport (OT):* The Benamou-Brenier equation (Benamou & Brenier, 2000) below provides a dynamic formulation of Optimal Transport $T$, which transports probability mass from $\pi_0$ to $q$ via a velocity field $v(x(t), t)$:

$$\inf_v \int_0^1 \mathbb{E}_{x(t) \sim \rho(\cdot, t)} \|v(x(t), t)\|^2 dt$$
$$\text{s.t.} \quad \partial_t \rho + \nabla \cdot (\rho v) = 0, \quad \rho(\cdot, 0) = \pi_0, \quad \rho(\cdot, 1) = q, \tag{3}$$

The optimization problem seeks to find the optimal $T$ that moves mass from the base density $\pi_0$ to the target density $q$, subject to the continuity equation (2) to ensure that $\rho(\cdot, t)$ evolves as a valid probability density over time. Additionally, the constraint $\rho(\cdot, 1) = q$ ensures that the target density is reached by the end of the time horizon. The time horizon is rescaled to $[0, 1]$ without loss of generality.

## 3. Annealing Flow Model

The annealing philosophy (Gelfand et al., 1990; Sorkin, 1991; Van Groenigen & Stein, 1998; Neal, 2001) refers to gradually transitioning an initial flattened distribution to the target distribution as the temperature decreases. Building on this idea, we introduce Annealing Flow (AF), a sampling algorithm that learns a continuous normalizing flow to gradually map an initial easy-to-sample density $\pi_0(x)$ to the target density $q(x)$ through a set of intermediate distributions.

We define the target distribution as $q(x) = Z\tilde{q}(x)$, where $\tilde{q}(x)$ is the unnormalized form given explicitly, and $Z > 0$ is an unknown normalizing constant. To interpolate between an easy-to-sample initial distribution $\pi_0(x)$ (e.g., a standard Gaussian) and the target $q(x)$, we construct a sequence of intermediate distributions $\{f_k(x)\}$, with corresponding unnormalized forms $\{\tilde{f}_k(x)\}$. These are defined as:

$$\tilde{f}_k(x) = \pi_0(x)^{1-\beta_k} \tilde{q}(x)^{\beta_k},$$
$$f_k(x) = Z_k \tilde{f}_k(x). \tag{4}$$

Here, $1/Z_k = \int_{\mathbb{R}^n} \tilde{f}_k(x)dx$ is an unknown normalizing constant, and $\beta_k$ is an increasing sequence with $\beta_0 = 0$ and $\beta_K = 1$. Therefore, $f_0(x) = \pi_0(x)$ and $f_K(x) = q(x)$. Importantly, our algorithm depends only on the unnormalized $\tilde{f}_k(x)$ and is independent of $Z_k$.

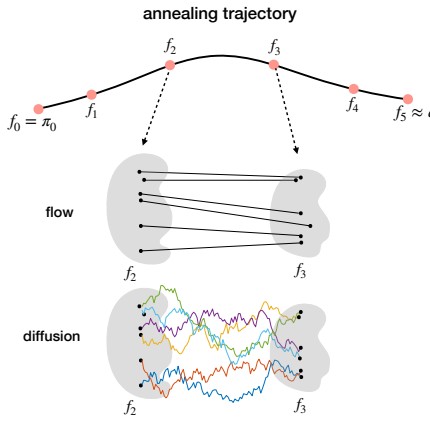

*Figure 1.* Illustrations and comparisons of annealing trajectories: Annealing Flow (based on optimal transport maps) versus other methods based on diffusion and score matching.

### 3.1. Algorithm

The objective of Annealing Flow (AF) is to learn a continuous optimal transport flow that transitions between intermediate distributions $f_{k-1}$ and $f_k$ for $k = 1, \ldots, K$, ultimately mapping an initial distribution $\pi_0(x)$ to the target distribution $q(x)$, as illustrated in Figure 1. A key observation is that during AF training, the normalizing constant does not affect the training objective. Once trained, users simply sample $x(0) \sim \pi_0(x)$, and the transport map pushes them to $x(1) \sim q(x)$. The transport map $T$ evolves the density according to (2), which in turn drives the evolution of the sample $x(t)$ following the ODE in (1):

$$x(t) = T(x(0)) = x(0) + \int_0^t v(x(s), s)ds, \quad t \in [0, 1].$$
(5)

We divide the time horizon $[0, 1]$ into $K$ intervals: $[t_{k-1}, t_k)$ for $k = 1, \ldots, K - 1$, and $[t_{K-1}, t_K]$, with $t_0 = 0$ and $t_K = 1$. We thus define $x(t_k)$ as the pushed-forward sample at time step $t_k$. Following the annealing flow path in (4), the continuous flow map $T$ gradually transforms the density from $f_0(x)$ to $f_1(x)$ over $[0, t_1)$, and continues this process until $f_{K-1}(x)$ is transformed into $f_K(x) = q(x)$ over $[t_{K-1}, 1]$.

Figure 2 shows this progression with two intermediate distributions. For clarity, we denote $T_k$ as the segment of the continuous normalizing flow during $[t_{k-1}, t_k)$, which pushes the density from $f_{k-1}(x)$ to $f_k(x)$. Consequently,

we define $v_k(x(t), t)$ as the velocity field of $v(x(t), t)$ during the time horizon $t \in [t_{k-1}, t_k)$.

Annealing Flow aims to learn each transport map $T_k$ based on dynamic OT objective (3) over the time horizon $[t_{k-1}, t_k)$, where the velocity field $v_k(x(t), t)$ is learned using a neural network. The terminal condition $\rho(\cdot, 1) = q$ in (3) can be imposed approximately using the Kullback–Leibler divergence (see, for instance, Ruthotto et al. (2020)). Consequently, minimizing the objective (3) for dynamic optimal transport $T_k : f_{k-1}(x) \to f_k(x)$ over the time horizon $[t_{k-1}, t_k)$ can be reduced to solving the following problem:

$$T_k = \arg\min_T \left\{ \text{KL}(T_\# f_{k-1} \| f_k) \right.$$
$$\left. + \gamma \int_{t_{k-1}}^{t_k} \mathbb{E}_{x(t) \sim \rho(\cdot, t)} \| v_k(x(t), t) \|^2 dt \right\},$$
(6)

$t \in [t_{k-1}, t_k)$, subject to $\rho(x(t), t)$ and $v_k(x(t), t)$ evolving according to (2). Here, $\text{KL}(T_\# f_{k-1} \| f_k)$ represents the KL divergence between the push-forward density $T_\# f_{k-1}$ (by transport map $T$) and the target density $f_k$, and $\gamma > 0$ is a regularization parameter that controls the smoothness and efficiency of the transport path. Additionally, the constraint (2) ensures that $x(t)$ follows the ODE trajectory defined by (1) during $t \in [t_{k-1}, t_k)$, which is given by:

$$x(t) = x(t_{k-1}) + \int_{t_{k-1}}^t v_k(x(s), s)ds, \quad t \in [t_{k-1}, t_k).$$
(7)

The following proposition shows that, once samples $x(t_{k-1})$ from $f_{k-1}$ have been obtained through the previously learned $T_1, \cdots, T_{k-1}$, the KL divergence in (6) can be equivalently expressed using $v_k(x(t), t)$ and $-\log \tilde{f}_k(x)$, up to an additive constant. Therefore, learning the optimal transport map $T_k$ reduces to learning the optimal $v_k(x(t), t)$. The proof is provided in Appendix A.1.

**Proposition 3.1.** *Given the samples from $f_{k-1}$, we have:*

$$\text{KL}(T_\# f_{k-1} \| f_k) = c + \mathbb{E}_{x(t_{k-1}) \sim f_{k-1}} \left[ -\log \tilde{f}_k(x(t_k)) \right.$$
$$\left. - \int_{t_{k-1}}^{t_k} \nabla \cdot v_k(x(s), s) \, ds \right],$$
(8)

*up to a constant $c$ that is independent of $v_k(x(s), s)$.*

Given $x(t_{k-1})$ from $f_{k-1}(x)$, the value of $x(t_k)$ inside $-\log \tilde{f}_k(\cdot)$ can be calculated as shown in equation (7). Additionally, according to the lemma below, the second term in the objective (6) can be approximated as a discretized sum. The proof is provided in Appendix A.1.

**Lemma 3.2.** *Let $x(t)$ be particle trajectories driven by a smooth velocity field $v_k(x(t), t)$ over the time interval*



$$T_0 \atop [0, t_1)$$

$$T_1 \atop [t_1, t_2)$$

$$T_2 \atop [t_2, 1]$$

(a) $\beta_0 = 0$      (b) $\beta_1 = 1/3$      (c) $\beta_2 = 2/3$      (d) $\beta_3 = 1$

*Figure 2.* Illustration of the Annealing Flow Map, with a set of annealing densities from $\pi_0(x) = N(0, I_2)$ to $q(x)$, a GMM with 6 modes.

$[t_{k-1}, t_k)$, where $h_k = t_k - t_{k-1}$. *Assume that $v_k(x, t)$ is Lipschitz continuous in both $x$ and $t$. By dividing $[t_{k-1}, t_k)$ into $S$ equal mini-intervals with grid points $t_{k-1,s}$, where $s = 0, 1, \ldots, S$ and $t_{k-1,0} = t_{k-1}$, $t_{k-1,S} = t_k$, we have:*

$$\int_{t_{k-1}}^{t_k} \mathbb{E}_{x(t) \sim \rho(\cdot, t)} \left[ \| v_k(x(t), t) \|^2 \right] dt$$
$$= \frac{S}{h_k} \sum_{s=0}^{S-1} \mathbb{E} \left[ \| x(t_{k-1,s+1}) - x(t_{k-1,s}) \|^2 \right] + O(h_k^2/S). \tag{9}$$

One can observe that the RHS of (9) is a discretized sum of stepwise Wasserstein-2 distances, reflecting the dynamic nature of the transport. Dynamic $W_2$ regularization encourages smooth transitions from $f_{k-1}$ to $f_k$ with minimal transport cost, ensuring stable sampling performance.

Next, by incorporating Proposition 3.1 and Lemma 3.2 into (6), the *final objective* becomes:

$$\min_{v_k(\cdot, t)} \mathbb{E}_{x(t_{k-1}) \sim f_{k-1}} \left[ - \log \tilde{f}_k(x(t_k))^\dagger - \int_{t_{k-1}}^{t_k} \nabla \cdot v_k(x(s), s) ds \right.$$
$$\left. + \alpha \sum_{s=0}^{S-1} \| x(t_{k-1,s+1}) - x(t_{k-1,s}) \|^2 \right]. \tag{10}$$

*Remark*[†]: We empirically observed that replacing the first term $- \log \tilde{f}_k(x(t_k))$ in the objective (10) with the first-order Taylor approximation $-h_k \nabla \log \tilde{f}_k(x(t_k)) \cdot v_k$ led to improved empirical performance, as detailed in Appendix C.1.

In the final objective, $\alpha = \gamma S / h_k$, and $v_k(x(s), s)$ is learned by a neural network. We break the time interval $[t_{k-1}, t_k)$ into $S$ mini-intervals, and $x(t_{k-1,s+1})$ is computed as in equation (7).

### 3.2. Properties of learned velocity field

As $h_k = t_k - t_{k-1} \to 0$, the final objective in (10) converges to a form involving the Stein operator. This is shown in A.2 as part of the prerequisite proof for Theorem 3.3.

Define $L^2(f_{k-1}) = \{ v : \int_{\mathbb{R}^d} \| v(x) \|^2 f_{k-1}(x) \, dx < \infty \}$ as the $L^2$ space over $(\mathbb{R}^d, f_{k-1}(x) \, dx)$. Define $s_k = \nabla \log f_k$ as the score function of the density $f_k$. We can then establish the following property, with proofs provided in A.2:

**Theorem 3.3.** (Optimal Velocity Field as Score Difference) *Suppose $h_k \to 0$. Let $f_{k-1}$ and $f_k$ be continuously differentiable on $\mathbb{R}^d$. Assume that $\nabla \cdot v_k(x)$ exists for all $x \in \mathbb{R}^d$, and $\nabla \cdot v_k(x)$, $s_{k-1}$ and $s_k$ belong to $L^2(f_{k-1})$. Assume that the components of $v_k$ are independent and $\lim_{\|x\| \to \infty} f_{k-1}(x) \| v_k(x) \|_2 = 0$. Under these conditions, the minimizer of (10) is:*

$$\lim_{h_k \to 0} v_k^* = s_k - s_{k-1}. \tag{11}$$

Therefore, the infinitesimal optimal $v_k^*$ is equal to the difference between the score function of the next density, $f_k$, and the current density, $f_{k-1}$. This suggests that by adding more intermediate densities, one can learn a sufficiently smooth transport map $T$ that exactly learns the mapping between each pair of densities.

Additionally, one can observe that when each $\tilde{f}_k(x)$ is set to the target $\tilde{q}(x)$, i.e., when all $\beta_k$ are set to 1, and the second term in the objective (6) is relaxed to static $W_2$ regularization, the objective of Annealing Flow becomes equivalent to Wasserstein gradient flow. Details, along with its convergence theorems, are provided in Appendix B.

## 4. Algorithm Implementation

---

**Algorithm 1** Block-wise Training of Annealing Flow Net

---

**Require:** Unnormalized target density $\tilde{q}(x)$; an easy-to-sample $\pi_0(x)$; $\{\beta_1, \beta_2, \cdots, \beta_{K-1}\}$; total number of blocks $K$.

1: Set $\beta_0 = 0$ and $\beta_K = 1$
2: **for** $k = 1$ to $K$ **do**
3:      Set $\tilde{f}_k(x) = \pi_0(x)^{1-\beta_k} \tilde{q}(x)^{\beta_k}$;
4:      Sample $\{x^{(i)}(0)\}_{i=1}^n$ from $\pi_0(x)$
5:      Compute the pushed-forward samples $x^{(i)}(t_{k-1})$ from the trained $(k-1)$ blocks via (13);
6:      *(Optional Langevin adjustments)*
7:          **for** several steps, update each $x^{(i)}(t_{k-1})$ via
8:          $x^{(i)}(t_{k-1}) \leftarrow x^{(i)}(t_{k-1}) + \frac{\eta}{2} \nabla \log \tilde{f}_{k-1} + \sqrt{\eta}\, \epsilon$
9:      **end for**
10:      Optimize $v_k(\cdot, t)$ by minimizing the final objective.
11: **end for**

---

## 4.1. Block-wise training

Training of the $k$-th flow map $T_k$ begins once the $(k-1)$-th block has completed training. Given the samples $\{x^{(i)}(t_{k-1})\}_{i=1}^n \sim f_{k-1}(x)$ after the $(k-1)$-th block, we can replace $\mathbb{E}_{x \sim f_{k-1}}$ with the empirical average. The divergence of the velocity field in (10) can be computed either by brute force or via the Hutchinson trace estimator (Hutchinson, 1989; Xu et al., 2024b):

$$\nabla \cdot v_k(x,t) \approx \mathbb{E}_{\epsilon \sim p_\epsilon}\left[\epsilon \cdot \frac{v_k(x+\sigma\epsilon,t)-v_k(x,t)}{\sigma}\right], \quad (12)$$

where we select $p(\epsilon) = N(0, I_d)$. The Hutchinson estimator significantly improves computational efficiency while maintaining good performance. More details are provided in C.2. Furthermore, we apply the Runge-Kutta method for numerical integration, with details provided in C.3.

Our method adopts block-wise training for the continuous normalizing flow, as summarized in Algorithm 1. After training each block, optional Langevin adjustments (Lines 6–9) can be applied to enhance performance. The block-wise training approach of Annealing Flow significantly reduces memory and computational requirements, as only one neural network is trained at a time.

## 4.2. Efficient sampling and methods comparisons

Once $T$ is learned, the sampling process of the target $q(x)$ becomes very efficient. Users simply sample $\{x^{(i)}(0)\}_{i=1}^n$ from $\pi_0(x)$, and then directly calculate $\{x^{(i)}(1)\}_{i=1}^n \sim q(x)$ through Annealing Flow nets ($k = 1, \cdots, K$):

$$x^{(i)}(t_k) = T_k(x^{(i)}(t_{k-1}))$$
$$= x^{(i)}(t_{k-1}) + \int_{t_{k-1}}^{t_k} v_k(x^{(i)}(s), s)ds. \quad (13)$$

MCMC struggles with poor performance on multi-modal distributions, long mixing times, and correlated samples, especially in high dimensions. In contrast, NFs push samples from $\pi_0(x)$ through a learned transport map, enabling faster and independent sampling. Although Annealing Flow requires more expensive pre-training, this can be done offline and only once. Once trained, AF samplers are highly efficient, generating 10,000 samples in 2.1 seconds on average in our 50D experiments.

Compared to recent NFs, our annealing procedure enables a more direct and efficient exploration of transport trajectories. The lower part of Figure 1 illustrates a single transport trajectory between two consecutive annealing densities. Unlike many recent annealing-like NFs (Tian et al., 2024; Fan et al.), which depend on score estimation and matching for training, our Annealing Flow is based on a dynamic OT objective, free from score matching, and uniquely supported by dynamic $W_2$ regularization. This unique loss facilitates optimal paths with notably fewer time steps. Detailed results, algorithmic analyses, and ablation studies are thoroughly presented in Section 6 and Appendix D.

## 5. Importance Flow

Sampling from complex distributions is fundamental, which can benefit tasks like Bayesian analysis and statistical physics. Here, we briefly discuss another aspect: using Annealing Flow to sample from the Least-Favorable-Distribution (LFD) and obtain a low-variance Importance Sampling (IS) estimator, referred to as Importance Flow.

### 5.1. Settings

Suppose we want to estimate $\mathbb{E}_{X \sim \pi_0(x)}[h(X)]$, which cannot be computed in closed form. A natural approach is to use Monte Carlo estimation by sampling $\{x_i\}_{i=1}^n$ from $\pi_0(x)$. However, if $x_i$ consistently falls in regions where $h(x)$ has extreme values, the estimator may exhibit high variance. For example, with $\pi_0(x) = N(0, I_d)$ and $h(x) = 1_{\|x\| \geq 6}$, almost no samples will satisfy $\|x\| \geq 6$, resulting in a zero estimate.

To address this situation, we can select an appropriate proposal distribution $q(x)$ and rewrite the expectation and MC estimator as:

$$\mathbb{E}_{x \sim \pi_0(x)}[h(x)] = \mathbb{E}_{x \sim q(x)}\left[\frac{\pi_0(x)}{q(x)}h(x)\right]$$
$$\approx \frac{1}{n}\sum_{i=1}^n \frac{\pi_0(x_i)}{q(x_i)}h(x_i), \ x_i \sim q(x). \quad (14)$$

It is well-known that the theoretically optimal proposal for the importance sampler is: $q^*(x) \propto \pi_0(x)|h(x)| := \tilde{q}^*(x)$. However, $\tilde{q}^*(x)$ is often difficult to sample from, especially when $\pi_0(x)$ or $h(x)$ is complex. Consequently, people typically choose a distribution that is similar in shape to the theoretically optimal proposal but easier to sample from.

Annealing Flow enables sampling from $q^*(x)$, allowing the construction of an Importance Sampling (IS) estimator. However, $q^*(x)$ is only known up to the normalizing constant $Z$, where $q^*(x) = \frac{1}{Z}\tilde{q}^*(x)$ and $Z = \mathbb{E}_{x \sim \pi_0(x)}[h(x)]$ is our target. Therefore, assuming no knowledge of $Z$, a common choice can be the Normalized IS Estimator:

$$\hat{I}_N = \sum_{i=1}^n \frac{\pi_0(x_i)}{\tilde{q}^*(x_i)}h(x_i)\Big/\sum_{i=1}^n \frac{\pi_0(x_i)}{\tilde{q}^*(x_i)}. \quad (15)$$

However, this estimator is often biased, as can be seen from Jensen's Inequality.

## 5.2. Density ratio estimation

Using samples from $q^*(x)$ and those along the trajectory obtained via Annealing Flow, we can train a neural network for Density Ratio Estimation (DRE) of $\pi_0(x)/q^*(x)$. Inspired by works Rhodes et al. (2020); Choi et al. (2022); Xu et al. (2025), we can train a continuous neural network

$$r(x) = r_K(x;\theta_K) \circ r_{K-1}(x;\theta_{K-1}) \circ \cdots \circ r_1(x;\theta_1), \quad (16)$$

where samples $x_i \sim f_K = q^*(x)$ are inputs and the output is the density ratio $\pi_0(x_i)/q^*(x_i)$. Each $r_k(x;\theta_k)$ is trained using the following loss:

$$\mathcal{L}_k(\theta_k) = \mathbb{E}_{x(t_{k-1})\sim f_{k-1}} \left[ \log(1 + e^{-r_k(x_i(t_{k-1}))}) \right]$$
$$+ \mathbb{E}_{x(t_k)\sim f_k} \left[ \log(1 + e^{r_k(x_i(t_k))}) \right]. \quad (17)$$

The theoretically optimal $r_k^*(x) = \log(f_{k-1}(x)/f_k(x))$, and thus $r^*(x) = \sum_{k=1}^K r_k^*(x) = \log(\pi_0(x)/q^*(x))$. (Appendices A.3 and C.5 contain proof and further details.) To obtain the optimal importance sampling estimator, one can then directly use samples $\{x_i\}_{i=1}^n \sim q^*(x)$ from Annealing Flow and apply (14) together with the DRE: $\frac{1}{n}\sum_{i=1}^n \exp(r^*(x_i)) \cdot h(x_i)$. The estimator is unbiased and can achieve zero variance theoretically.

# 6. Numerical Experiments

In this section, we present numerical experiments comparing Annealing Flow (AF) with the following methods: (1) Annealing-based MCMC: Parallel Tempering (PT); (2) NN-assisted MCMC: AI-Sampler (AIS) (Egorov et al., 2024); (3) Particle Optimization methods: Stein Variational Gradient Descent (SVGD) (Liu & Wang, 2016) and Mollified Interaction Energy Descent (MIED) (Li et al., 2023); and (4) Normalizing Flow approaches: Continual Repeated Annealed Flow Transport Monte Carlo (CRAFT) (Matthews et al., 2022), Liouville Flow Importance Sampler (LFIS) (Tian et al., 2024), and Path-Guided Particle-based Sampling (PGPS) (Fan et al.). Experimental details are provided in Appendix C.3.

The number of time steps for Annealing Flow (AF) is set per distribution as follows: (1) GMMs: 12 steps—10 intermediate densities and 2 refinement blocks using $q(x)$ as the target; (2) Truncated Normal: 8 steps with 7 intermediate densities; (3) Funnel: 8 steps with each intermediate density as the target; (4) Exp-Weighted Gaussian: 20 steps—15 intermediate densities and 5 refinement blocks; (5) Bayesian Logistic Regression: 6 steps with each intermediate density as the target.

For fair comparison, all figures and tables (except Table 3)

Our code is publicly available at https://github.com/StatFusion/Annealing-Flow-For-Sampling.

use the same number of training steps and intermediate densities for CRAFT, LFIS, PGPS, and our AF, with identical network architectures and training iterations. All AF results are reported without the optional Langevin adjustments, and PGPS is also run without them for fairness. Additional results—including LFIS with 256 steps, and CRAFT/PGPS with 128 steps—are provided in Appendix D.

## 6.1. Experiments and results

*Gaussian Mixture Models (GMMs):* Figure 3 (main script) and Figures 4 and 5 (Appendix) show the visualized sampling results of various methods on GMMs. Additionally, we tested unequally weighted GMMs, where two modes have double the weight of the others. Numerical evaluation metrics for experiments with different numbers of modes and dimensions are presented in Tables 1, 4, and 5. Additional figures and tables are provided in Appendix D.

*Truncated normal distribution:* Figure 7 in the Appendix shows the sampling results for the truncated normal distribution $\tilde{q}(x) = 1_{\|x\|\geq c}N(0, I_d)$. Notably, even after relaxing $1_{\|x\|>c}$ to $1/(1 + \exp(-k(\|x\| - c)))$ in $\nabla \log (1_{\|x\|\geq c}N(0, I_d))$, methods including CRAFT, LFIS, PGPS, SVGD, MIED, and AI-Sampler fail to produce meaningful results. Please refer to Figure 6 in Appendix D for the results of these algorithms.

*Funnel distribution:* We tested each algorithm for the funnel distribution

$$q(x_1, x_2, \ldots, x_d) \propto \mathcal{N}(x_1 \mid 0, \sigma^2) \prod_{i=2}^d \mathcal{N}(x_i \mid 0, \exp(x_1)). \quad (18)$$

Figure 9 in Appendix D shows the sampling results.

*Exp-Weighted Gaussian (ExpGauss) with an extreme number of modes in high-dimensional spaces:*

We tested each algorithm on sampling from an extreme distribution:

$$p(x_1, x_2, \cdots, x_{10}) \propto e^{10\sum_{i=1}^{10}|x_i| - \frac{1}{2}\|x\|^2}, \quad (19)$$

which has $2^{10} = 1024$ modes arranged at the vertices of a *10-D* cube. The Euclidean distance between two horizontally or vertically adjacent modes is 20, while the diagonal modes are separated by up to $\sqrt{10 \cdot 20^2} \approx 63.25$. We also tested on the extreme distribution:

$$p(x_1, x_2, \cdots, x_{50}) \propto e^{10\sum_{i=1}^{10}\frac{|x_i|}{\sigma_i^2} + 10\sum_{i=11}^{50}\frac{x_i}{\sigma_i^2} - \frac{1}{2}\|x\|^2}, \quad (20)$$

which has $2^{10} = 1024$ modes arranged at the vertices of a 50-dimensional space, with imbalanced variances across modes along different dimensions.

Given the challenge of visualizing results in high-dimensional space, we first present the number of modes

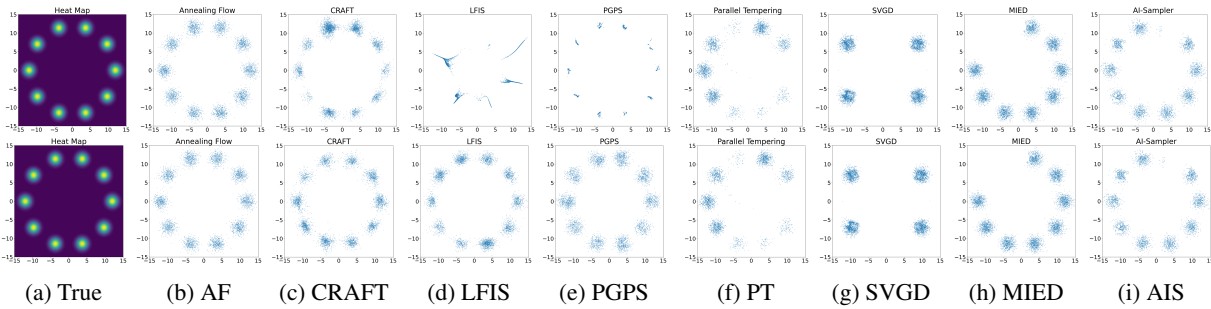

|  | (a) True | (b) AF | (c) CRAFT | (d) LFIS | (e) PGPS | (f) PT | (g) SVGD | (h) MIED | (i) AIS |

*Figure 3.* Sampling from a 10-mode Gaussian Mixture Model (GMM) arranged in a circle of radius $r = 12$. Annealing Flow (AF) always uses 12 time steps. Top row: AF, CRAFT, LFIS, PGPS are trained using 12 time steps; PGPS is trained without Langevin correction. Bottom row: AF, CRAFT, LFIS, and PGPS are trained using 12, 128, 256, and 128 time steps, respectively; PGPS is trained with Langevin correction. See Figures 4 and 5 for additional visualizations.

*Table 1.* Mode-Weight Mean Squared Error across distributions. The number of time steps for CRAFT, LFIS, and PGPS is set equal to that of AF. Three additional types of metrics are presented in Tables 4, 5, and 7.

|  | Distributions | AF | CRAFT | LFIS | PGPS | SVGD | MIED | AI-Sampler |
|---|---|---|---|---|---|---|---|---|
| $d = 2$ | GMM-6-8 | $(\mathbf{8.5\pm_{.37}}) \times \mathbf{10^{-5}}$ | $(\mathbf{7.7\pm_{.51}}) \times \mathbf{10^{-5}}$ | $(1.2\pm_{.28}) \times 10^{-4}$ | $(\mathbf{7.4\pm_{.44}}) \times \mathbf{10^{-5}}$ | $(1.6\pm_{.84}) \times 10^{-2}$ | $(1.4\pm_{.79}) \times 10^{-3}$ | $(\mathbf{8.3\pm_{.48}}) \times \mathbf{10^{-5}}$ |
|  | GMM-8-10 | $(\mathbf{9.4\pm_{.12}}) \times \mathbf{10^{-5}}$ | $(2.7\pm_{.39}) \times 10^{-4}$ | $(5.8\pm_{.82}) \times 10^{-4}$ | $(\mathbf{9.8\pm_{.41}}) \times \mathbf{10^{-5}}$ | $(1.7\pm_{.92}) \times 10^{-2}$ | $(1.5\pm_{.87}) \times 10^{-3}$ | $(2.3\pm_{.52}) \times 10^{-4}$ |
|  | GMM-10-12 | $(\mathbf{5.7\pm_{.76}}) \times \mathbf{10^{-5}}$ | $(2.6\pm_{.70}) \times 10^{-4}$ | $(2.3\pm_{.41}) \times 10^{-3}$ | $(2.2\pm_{.82}) \times 10^{-4}$ | $(1.7\pm_{.82}) \times 10^{-2}$ | $(5.6\pm_{.77}) \times 10^{-3}$ | $(1.0\pm_{.68}) \times 10^{-3}$ |
|  | wGMM-10-12 | $(\mathbf{9.5\pm_{.70}}) \times \mathbf{10^{-5}}$ | $(\mathbf{9.7\pm_{.98}}) \times \mathbf{10^{-5}}$ | $(3.8\pm_{.62}) \times 10^{-3}$ | $(4.7\pm_{.85}) \times 10^{-4}$ | $(2.4\pm_{.93}) \times 10^{-2}$ | $(5.2\pm_{.60}) \times 10^{-3}$ | $(3.6\pm_{.60}) \times 10^{-3}$ |
| $d = 5$ | GMM-6-8 | $(\mathbf{1.3\pm_{.48}}) \times \mathbf{10^{-4}}$ | $(1.5\pm_{.18}) \times 10^{-2}$ | $(5.4\pm_{.13}) \times 10^{-4}$ | $(\mathbf{1.2\pm_{.57}}) \times \mathbf{10^{-4}}$ | $(2.1\pm_{.75}) \times 10^{-2}$ | $(1.1\pm_{.40}) \times 10^{-3}$ | $(8.6\pm_{.21}) \times 10^{-3}$ |
|  | GMM-8-10 | $(\mathbf{2.2\pm_{.29}}) \times \mathbf{10^{-4}}$ | $(1.1\pm_{.79}) \times 10^{-2}$ | $(9.2\pm_{.41}) \times 10^{-4}$ | $(4.8\pm_{.85}) \times 10^{-4}$ | $(3.4\pm_{.75}) \times 10^{-2}$ | $(4.8\pm_{.94}) \times 10^{-3}$ | $(2.7\pm_{.65}) \times 10^{-3}$ |
|  | GMM-10-12 | $(\mathbf{1.8\pm_{.65}}) \times \mathbf{10^{-4}}$ | $(8.8\pm_{.79}) \times 10^{-3}$ | $(3.5\pm_{.49}) \times 10^{-3}$ | $(5.2\pm_{.73}) \times 10^{-4}$ | $(5.2\pm_{1.8}) \times 10^{-2}$ | $(3.1\pm_{.14}) \times 10^{-3}$ | $(4.5\pm_{.80}) \times 10^{-3}$ |
|  | wGMM-10-12 | $(\mathbf{7.3\pm_{.76}}) \times \mathbf{10^{-4}}$ | $(9.7\pm_{.29}) \times 10^{-3}$ | $(7.6\pm_{.71}) \times 10^{-3}$ | $(5.9\pm_{.78}) \times 10^{-4}$ | $(6.4\pm_{.39}) \times 10^{-2}$ | $(6.0\pm_{.65}) \times 10^{-3}$ | $(7.9\pm_{.32}) \times 10^{-3}$ |
| $d = 10$ | ExpGauss-1024 | $(8.2\pm_{.31}) \times 10^{-8}$ | $(8.8\pm_{.35}) \times 10^{-7}$ | $(1.5\pm_{.66}) \times 10^{-6}$ | $(1.2\pm_{.35}) \times 10^{-7}$ | $(\mathbf{8.5\pm_{.27}}) \times \mathbf{10^{-8}}$ | $(4.6\pm_{.18}) \times 10^{-6}$ | $(8.6\pm_{.24}) \times 10^{-5}$ |
| $d = 50$ | ExpGauss-1024 | $(\mathbf{9.8\pm_{.68}}) \times \mathbf{10^{-8}}$ | $(9.4\pm_{.80}) \times 10^{-7}$ | $(2.2\pm_{.82}) \times 10^{-6}$ | $(3.5\pm_{.80}) \times 10^{-7}$ | $(3.3\pm_{1.0}) \times 10^{-6}$ | $(2.1\pm_{.92}) \times 10^{-6}$ | $(1.1\pm_{.76}) \times 10^{-4}$ |
|  | ExpGaussUV-2-1024 | $(\mathbf{1.8\pm_{.97}}) \times \mathbf{10^{-7}}$ | $(7.8\pm_{.59}) \times 10^{-6}$ | $(7.1\pm_{.23}) \times 10^{-6}$ | $(4.8\pm_{.23}) \times 10^{-6}$ | $(7.6\pm_{.97}) \times 10^{-6}$ | $(9.2\pm_{1.1}) \times 10^{-6}$ | $(1.7\pm_{.95}) \times 10^{-4}$ |
|  | ExpGaussUV-10-1024 | $(\mathbf{2.4\pm_{.13}}) \times \mathbf{10^{-7}}$ | $(1.9\pm_{.98}) \times 10^{-5}$ | $(6.9\pm_{.47}) \times 10^{-5}$ | $(9.9\pm_{.70}) \times 10^{-6}$ | $(1.1\pm_{.98}) \times 10^{-5}$ | $(9.5\pm_{1.9}) \times 10^{-6}$ | $(1.4\pm_{.99}) \times 10^{-4}$ |
|  | ExpGaussUV-15-1024 | $(\mathbf{2.0\pm_{.31}}) \times \mathbf{10^{-7}}$ | $(2.2\pm_{1.1}) \times 10^{-5}$ | $(8.9\pm_{.66}) \times 10^{-5}$ | $(1.9\pm_{.45}) \times 10^{-5}$ | $(7.0\pm_{.66}) \times 10^{-5}$ | $(3.7\pm_{.59}) \times 10^{-5}$ | $(1.6\pm_{.87}) \times 10^{-4}$ |
|  | ExpGaussUV-20-1024 | $(\mathbf{3.6\pm_{.60}}) \times \mathbf{10^{-7}}$ | $(4.7\pm_{.78}) \times 10^{-5}$ | $(9.7\pm_{.94}) \times 10^{-5}$ | $(4.8\pm_{.82}) \times 10^{-5}$ | $(8.7\pm_{.93}) \times 10^{-5}$ | $(7.6\pm_{.71}) \times 10^{-5}$ | $(1.8\pm_{.90}) \times 10^{-4}$ |

*Table 2.* Number of modes explored by different methods in a 50-dimensional Exponentially Weighted Gaussian with 1024 far-separated modes. The number of time steps for AF, CRAFT, LFIS, and PGPS is set to 20.

|  | True | AF | CRAFT | LFIS | PGPS | PT | SVGD | MIED | AIS |
|---|---|---|---|---|---|---|---|---|---|
| $d = 2$ | 4 | **4** | 2.6 | 3.6 | 3.8 | 3.4 | 3.9 | 3.8 | 3.8 |
| $d = 5$ | 32 | **32** | 22.3 | 27.4 | 31.2 | 25.2 | 28.5 | 28.0 | 28.3 |
| $d = 10$ | 1024 | **1024** | 515.4 | 387.0 | 826.0 | 233.7 | 957.3 | 923.4 | 301.2 |
| $d = 50$ | 1024 | **1024** | 473.2 | 298.2 | 813.6 | < 10 | 916.4 | 890.6 | 125.6 |

successfully explored by different algorithms across varying dimensions in Table 2. Each algorithm was run 10 times, with 20,000 points sampled per run, and the average number of modes explored was calculated. Additionally, Table 1 presents the Mode Weights Mean Squared Error along with results of other distributions.

*Evaluation metrics:* We report the following metrics for each applicable experiment: (1) Mode-Weight Mean Squared Error, (2) Maximum Mean Discrepancy (MMD), (3) Wasserstein Distance, and (4) Sample-Variance Mean Squared Error across dimensions. The results for these metrics are presented in Table 1 (main script) and in Tables 4, 5 and 7 in the Appendix. Appendix C.7 explains

the details of these metrics. In the tables, GMM refers to Gaussian Mixture Models, while wGMM denotes unequally weighted GMM. The notation (w)GMM-{number of modes}-{radius of the circle} represents a (w)GMM with the specified number of modes arranged on a circle of the given radius. ExpGauss-1024 refers to exponentially weighted Gaussian experiments involving 1,024 widely separated modes. ExpGaussUV-{number of UV dimensions}-1024 refers to ExpGauss with 1024 modes that have Unequal Variances (UV) across the specified number of dimensions. Alongside Table 7, we provide more results for ExpGauss, with details on the selection of $\sigma_i^2$ across dimensions.

*Bayesian logistic regression:* We use the same Bayesian logistic regression setting as in Liu & Wang (2016), where a hierarchical structure is assigned to the model parameters. The weights $\beta$ follow a Gaussian prior $p_0(\beta|\alpha) = N(\beta; 0, \alpha^{-1})$, and $\alpha$ follow a Gamma prior $p_0(\alpha) = \text{Gamma}(\alpha; 1, 0.01)$. Sampling is performed on the posterior $p(\beta, \alpha|D)$, where $D = \{x_i, y_i\}_{i=1}^n$. The performance comparisons are shown in Table 3. Detailed settings are given in C.4.

*Importance flow:* Table 8 in the Appendix reports the preliminary results of the importance flow (discussed in Section 5) for estimating $\mathbb{E}_{x \sim N(0,I)}\left[\mathbb{1}_{\|x\| \geq c}\right]$ with varying radii $c$ and

*Table 3.* Bayesian logistic regression: comparison of different algorithms across datasets. In the table $Acc_{\pm Std}$ represents Accuracy (%) $\pm$ std (%), and $l_{post}$ represents the averaged Log-Posterior. The number of time steps is set to 6, 128, 256, and 128 for AF, CRAFT, LFIS, and PGPS, respectively.

| Dataset \ Methods | AF | | CRAFT | | LFIS | | PGPS | | SVGD | | MIED | | AI-Sampler | |
|---|---|---|---|---|---|---|---|---|---|---|---|---|---|---|
| | $Acc_{\pm Std}$ | $l_{post}$ | $Acc_{\pm Std}$ | $l_{post}$ | $Acc_{\pm Std}$ | $l_{post}$ | $Acc_{\pm Std}$ | $l_{post}$ | $Acc_{\pm Std}$ | $l_{post}$ | $Acc_{\pm Std}$ | $l_{post}$ | $Acc_{\pm Std}$ | $l_{post}$ |
| Diabetes ($d=8$) | **76.3**$_{\pm 2.1}$ | **−0.50** | 76.2$_{\pm 2.2}$ | −0.51 | **76.2**$_{\pm 3.0}$ | **−0.50** | **76.3**$_{\pm 2.2}$ | **−0.50** | 76.1$_{\pm 2.5}$ | −0.50 | 75.8$_{\pm 2.3}$ | −0.50 | **76.3**$_{\pm 2.2}$ | **−0.49** |
| Cancer ($d=10$) | 97.9$_{\pm 1.1}$ | −0.02 | **98.9**$_{\pm 1.2}$ | −0.01 | 96.4$_{\pm 2.0}$ | −0.03 | **98.9**$_{\pm 1.1}$ | −0.01 | 97.8$_{\pm 2.5}$ | −0.02 | **98.9**$_{\pm 1.0}$ | **−0.01** | 97.8$_{\pm 2.8}$ | −0.02 |
| Heart ($d=13$) | **88.5**$_{\pm 2.7}$ | **−0.32** | 88.4$_{\pm 3.0}$ | −0.32 | 86.6$_{\pm 3.6}$ | −0.43 | 87.2$_{\pm 2.9}$ | −0.40 | 79.4$_{\pm 3.8}$ | −0.59 | 86.7$_{\pm 2.2}$ | −0.32 | 84.2$_{\pm 2.5}$ | −0.46 |
| Australian ($d=14$) | **86.6**$_{\pm 1.2}$ | **−0.36** | 85.0$_{\pm 3.0}$ | −0.39 | 84.1$_{\pm 2.0}$ | −0.36 | 85.4$_{\pm 1.7}$ | −0.39 | 84.6$_{\pm 2.9}$ | −0.37 | 85.2$_{\pm 1.3}$ | −0.37 | 84.6$_{\pm 2.3}$ | −0.38 |
| Ijcnn1 ($d=22$) | **92.0**$_{\pm 0.1}$ | **−0.20** | 88.8$_{\pm 0.1}$ | −0.23 | 89.8$_{\pm 0.4}$ | −0.31 | 91.2$_{\pm 1.3}$ | −0.20 | 89.4$_{\pm 0.3}$ | −0.21 | 91.8$_{\pm 0.2}$ | −0.20 | 88.3$_{\pm 0.3}$ | −0.33 |
| Svmguide3 ($d=22$) | 80.0$_{\pm 1.0}$ | −0.47 | 78.6$_{\pm 0.9}$ | −0.50 | **80.3**$_{\pm 1.2}$ | −0.48 | 80.0$_{\pm 1.0}$ | −0.49 | 78.9$_{\pm 1.2}$ | −0.48 | **80.6**$_{\pm 1.0}$ | **−0.47** | 80.1$_{\pm 1.0}$ | −0.47 |
| German ($d=24$) | **78.0**$_{\pm 1.7}$ | **−0.47** | 77.5$_{\pm 1.7}$ | −0.48 | 76.7$_{\pm 2.3}$ | −0.49 | 77.7$_{\pm 1.6}$ | −0.48 | 76.4$_{\pm 1.7}$ | −0.48 | 77.2$_{\pm 1.8}$ | −0.48 | 76.9$_{\pm 1.8}$ | −0.48 |
| Splice ($d=61$) | **86.9**$_{\pm 1.7}$ | **−0.41** | 81.1$_{\pm 2.2}$ | −0.49 | 82.0$_{\pm 1.3}$ | −0.48 | 82.8$_{\pm 2.1}$ | −0.47 | 82.4$_{\pm 2.0}$ | −0.47 | 83.1$_{\pm 1.5}$ | −0.46 | 80.1$_{\pm 1.9}$ | −0.50 |

dimensions. Please refer to C.5 for detailed experimental settings. Additionally, we discussed a possible extension of the Importance Flow framework in D.4.

### 6.2. Results discussions and comparisons

*Significance of Annealing Procedures:* We comment that annealing procedures play a crucial role in the success of our AF in high-dimensional settings with widely separated modes. Appendix D.3.1 and D.3.2 present ablation studies for cases with no or very few annealing steps. This unique feature allows AF to succeed on challenging distributions, unlike most NFs used for sampling.

*Little dependence on Monte Carlo sampling:* During implementations, we observe that CRAFT heavily depends on the choice of the MC kernel, one of its major functioning components. LFIS depends on accurate score estimation for training, necessitating many more time steps. PGPS relies heavily on Langevin adjustments after each step, without which its performance becomes poor. In contrast, AF achieves success with minimal assistance.

*Computational and storage efficiency:* In Tables 9 and 10 of Appendix D, we report the training and sampling times for AF, CRAFT, LFIS, and PGPS. Notably, AF requires significantly fewer intermediate time steps than other normalizing flow methods while achieving superior performance in our experiments. We also provide ablation studies in D.3.2 with even fewer time steps. In addition, in our experiments, AF with a single 32-unit hidden layer and sigmoid activation suffices for most tasks, except Exp-weighted Gaussian, which requires only 32-32 hidden layers. This also yields notable storage savings—for instance, 2.1MB total for 20 networks trained in the 50D ExpGauss experiment.

*Training stability:* Our unique dynamic optimal transport (OT) objective with $W_2$ regularization is key to ensuring much more stable performance, even with far fewer intermediate time steps. In Appendix D.3.1 and D.3.2, we provide ablation studies highlighting the significance of our OT objective. Unlike the score-matching training, the unique dynamic OT objective of AF is crucial for ensuring stability,

achieving much higher efficiency, and enabling successful sampling in high-dimensional and multi-modal settings.

*Comparisons with MCMC*: Compared to MCMC methods, AF offers three key advantages. First, AF enables efficient sampling after a one-time offline training phase, whereas MCMC requires no training but suffers from long mixing times and high runtime per effective sample. Second, AF performs reliably in high-dimensional, multimodal settings where MCMC often fails to explore all modes, as evidenced by its poor coverage in the 50D ExpGauss experiment (Table 2). Third, AF generates balanced and independent samples by construction, while MCMC frequently produces imbalanced mode visitation due to stochastic mode-hopping.

## 7. Conclusions

In this paper, we have proposed the Annealing Flow (AF) algorithm, a novel approach for sampling from high-dimensional and multi-modal distributions. With the unique Annealing-guided dynamic optimal transport objective, AF offers multiple advantages over existing methods, including superior performance on extreme distributions, greatly improved training efficiency compared to other NF methods, minimal reliance on MC assistance, and enhanced training stability compared to other NF methods.

We establish in Theorem 3.3 that the infinitesimal optimal velocity field corresponds to the score difference between consecutive annealing densities, a property unique to our AF. This desirable feature sets our AF's objective apart from recent annealing-like methods (Tian et al., 2024; Fan et al.). In Appendix B, we demonstrate the equivalence of the AF objective with the Wasserstein gradient flow and its associated convergence theorems. A range of experiments demonstrate that AF performs well across a variety of challenging distributions and real-world datasets. Finally, the importance flow discussed in Section 5 may be extended to a distribution-free model, allowing one to learn an importance flow from a dataset for sampling its Least-Favorable Distribution (LFD) with minimal variance, as further discussed in D.4.

## Acknowledgment

This work is partially supported by an NSF DMS-2134037, CMMI-2112533, and the Coca-Cola Foundation. This work is also partially supported by the Stewart Fellowship.

## Impact Statement

Our work contributes to statistical sampling and machine learning, offering both theoretical and practical advances. Sampling is a fundamental and ubiquitous problem with broad applications in statistical physics, Bayesian modeling, and machine learning. While our method has broad potential societal implications, none of which we feel must be specifically highlighted here.

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

## A. Proofs

### A.1. Proofs in Section 3.1

**Proposition 3.1.** (KL-Divergence Decomposition) *Given the samples from $f_{k-1}$, the KL-Divergence between $T_\# f_{k-1}$ and $f_k$ is equivalent to:*

$$\mathrm{KL}(T_\# f_{k-1} \| f_k) = c + \mathbb{E}_{x \sim f_{k-1}} \left[ -\log \tilde{f}_k(x(t_k)) - \int_{t_{k-1}}^{t_k} \nabla \cdot v_k(x(s), s) \, ds \right],$$

*up to a constant $c$ that is independent of $v_k(x(s), s)$.*

*Proof:*

Let $\rho(x, t)$ denote the density evolution under the transport map $T$, as defined in the continuity equation (2). By the constraints in the transport map objective (3), we have $T_\# f_{k-1}(x) = \rho(x, t_k)$. The expression for KL-divergence is given by:

$$\mathrm{KL}(T_\# f_{k-1} \| f_k) = \mathbb{E}_{x \sim \rho(x, t_k)} \left[ \log \frac{T_\# f_{k-1}(x)}{f_k(x)} \right] = \mathbb{E}_{x \sim \rho(x, t_k)} \left[ \log T_\# f_{k-1}(x) - \log f_k(x) \right]. \tag{21}$$

Recall that in equation (4), we express $f_k(x) = Z_k \tilde{f}_k(x)$. Substituting this into the equation yields:

$$\mathrm{KL}(T_\# f_{k-1} \| f_k) = \mathbb{E}_{x \sim \rho(x, t_k)} \left[ \log T_\# f_{k-1}(x) - \log \tilde{f}_k(x) \right] - \log Z_k \tag{22}$$

$$= \mathbb{E}_{x \sim \rho(x, t_{k-1})} \left[ \log T_\# f_{k-1}(x(t_k)) - \log \tilde{f}_k(x(t_k)) \right] - \log Z_k, \tag{23}$$

where the second equality holds because the flow $T$ maps samples $x(t_{k-1})$ to $x(t_k)$ in a way that preserves the probability measure by the continuity equation (2). That is,

$$\mathbb{E}_{x \sim \rho(x, t_k)}[\cdot] = \mathbb{E}_{x \sim \rho(x, t_{k-1})} \left[ \cdot \big|_{x(t_{k-1}) \to x(t_k)} \right], \tag{24}$$

where the particles $x(t)$ evolve according to the Neural ODE (1).

Next, to compute $\log T_\# f_{k-1}(x(t_k))$, we use the fact that the dynamics of the pushforward density $\rho$ are governed by the velocity field $v_k(x(s), s)$:

$$\frac{d}{ds} \log \rho(x(s), s) = \frac{\nabla \rho(x(s), s) \cdot \partial_s x(s) + \partial_s \rho(x(s), s)}{\rho(x(s), s)} \tag{25}$$

$$= \frac{\nabla \rho \cdot v_k - \nabla \cdot (\rho v_k)}{\rho} \bigg|_{(x(s), s)} \quad \text{(by (1) and (2))} \tag{26}$$

$$= \frac{\nabla \rho \cdot v_k - (\nabla \rho \cdot v_k + \rho \nabla \cdot v_k)}{\rho} \bigg|_{(x(s), s)} \tag{27}$$

$$= -\nabla \cdot v_k(x(s), s). \tag{28}$$

Integrating this equation over the interval $s \in [t_{k-1}, t_k)$, we find:

$$\log T_\# f_{k-1}(x(t_k)) = \log \rho(x(t_k), t_k) = \log \rho(x(t_{k-1}), t_{k-1}) - \int_{t_{k-1}}^{t_k} \nabla \cdot v_k(x(s), s) ds. \tag{29}$$

We now substitute this result back into the KL-divergence expression:

$$\mathrm{KL}(T_\# f_{k-1} \| f_k) = \mathbb{E}_{x \sim \rho(x, t_{k-1})} \left[ \log \rho(x(t_{k-1}), t_{k-1}) - \int_{t_{k-1}}^{t_k} \nabla \cdot v_k(x(s), s) ds - \log \tilde{f}_k(x(t_k)) \right] - \log Z_k. \tag{30}$$

Note that $\mathbb{E}_{x \sim \rho(x(t_{k-1}), t_{k-1})} \left[ \log \rho(x(t_{k-1}), t_{k-1}) \right]$ is independent of $v_k(x(s), s) : t_{k-1} \to t_k$ and thus acts as a constant term, along with $-\log Z_k$, which we now denote as $c$:

$$c = \mathbb{E}_{x \sim \rho(x(t_{k-1}), t_{k-1})} \left[ \log \rho(x(t_{k-1}), t_{k-1}) \right] - \log Z_k. \tag{31}$$

Besides, after successfully training the previous velocity fields, we have $\rho(x(t_{k-1}), t_{k-1}) = f_{k-1}(x)$. Therefore, the relevant terms for the KL-divergence are:

$$\text{KL}(T_\# f_{k-1} \parallel f_k) = c + \mathbb{E}_{x \sim f_{k-1}} \left[ -\log \tilde{f}_k(x(t_k)) - \int_{t_{k-1}}^{t_k} \nabla \cdot v_k(x(s), s) ds \right]. \tag{32}$$

**Lemma 3.2.** (Wasserstein Distance Discretization) *Let $x(t)$ be particle trajectories driven by a smooth velocity field $v_k(x(t), t)$ over the time interval $[t_{k-1}, t_k)$, where $h_k = t_k - t_{k-1}$. Assume that $v_k(x, t)$ is Lipschitz continuous in both $x$ and $t$. By dividing $[t_{k-1}, t_k)$ into $S$ equal mini-intervals with grid points $t_{k-1,s}$, where $s = 0, 1, \ldots, S$ and $t_{k-1,0} = t_{k-1}$, $t_{k-1,S} = t_k$, we have:*

$$\int_{t_{k-1}}^{t_k} \mathbb{E}_{x(t) \sim \rho(\cdot, t)} \left[ \|v_k(x(t), t)\|^2 \right] dt = \frac{S}{h_k} \sum_{s=0}^{S-1} \mathbb{E} \left[ \|x(t_{k-1,s+1}) - x(t_{k-1,s})\|^2 \right] + O\left( h_k^2 / S \right).$$

*Proof:*

Consider particle trajectories $x(t)$ driven by a sufficiently smooth velocity field $v_k(x(t), t)$ over the time interval $[t_{k-1}, t_k)$, where $h_k = t_k - t_{k-1}$. We divide this interval into $S$ equal mini-intervals of length $\delta t = \frac{h_k}{S}$, resulting in grid points $t_{k-1,s} = t_{k-1} + s\delta t$ for $s = 0, 1, \ldots, S$, where $\delta t = \frac{t_k - t_{k-1}}{S}$.

Within each mini-interval $[t_{k-1,s}, t_{k-1,s+1}]$, we perform a Taylor expansion of $x(t)$ around $t_{k-1,s}$:

$$x(t_{k-1,s+1}) = x(t_{k-1,s}) + v_k(x(t_{k-1,s}), t_{k-1,s})\delta t + \frac{1}{2} \frac{dv_k}{dt} \delta t^2 + O(\delta t^3), \tag{33}$$

where $\frac{dv_k}{dt}$ denotes the total derivative of $v_k$ with respect to time.

The squared displacement over the mini-interval $[t_{k-1,s}, t_{k-1,s+1}]$ is given by:

$$\|x(t_{k-1,s+1}) - x(t_{k-1,s})\|^2 = \left\| v_k(x(t_{k-1,s}), t_{k-1,s})\delta t + \frac{1}{2} \frac{dv_k}{dt} \delta t^2 + O(\delta t^3) \right\|^2 \tag{34}$$

$$= \|v_k(x(t_{k-1,s}), t_{k-1,s})\|^2 \delta t^2 + O(\delta t^3), \tag{35}$$

as we assume that $v_k$ is $L$-Lipschitz continuous and it follows that $|\frac{dv_k}{dt}| \leq L$. The higher-order terms $O(\delta t^3)$ become negligible as $\delta t \to 0$.

Summing the expected squared displacements over all mini-intervals, we obtain:

$$\sum_{s=0}^{S-1} \mathbb{E} \left[ \|x(t_{k-1,s+1}) - x(t_{k-1,s})\|^2 \right] = \delta t^2 \sum_{s=0}^{S-1} \mathbb{E} \left[ \|v_k(x(t_{k-1,s}), t_{k-1,s})\|^2 \right] + O\left( S \cdot \delta t^3 \right). \tag{36}$$

Now, we examine the L.H.S. of Lemma 3.2 by approximating the integral of the expected squared velocity using a Riemann sum:

$$\int_{t_{k-1}}^{t_k} \mathbb{E}_{x(t)} \left[ \|v_k(x(t), t)\|^2 \right] dt = \delta t \sum_{s=0}^{S-1} \mathbb{E} \left[ \|v_k(x(t_{k-1,s}), t_{k-1,s})\|^2 \right] + O\left( S \cdot \delta t^2 \right)$$

$$= \delta t \left[ \frac{1}{\delta t^2} \sum_{s=0}^{S-1} \mathbb{E} \left[ \|x(t_{k-1,s+1}) - x(t_{k-1,s})\|^2 \right] + O(S \cdot \delta t) \right] + O(S \cdot \delta t^2) \tag{37}$$

$$= \frac{1}{\delta t} \sum_{s=0}^{S-1} \mathbb{E} \left[ \|x(t_{k-1,s+1}) - x(t_{k-1,s})\|^2 \right] + O\left( S \cdot \delta t^2 \right),$$

where the Riemann sum error term $O(S \cdot \delta t^2)$ arises from a well-known result (for instance, see Chapter 1 of Axler (2020)), given the assumption that $v_k$ is $L-$Lipschitz continuous.

## A.2. Proofs in Section 3.2

Before formally proving Theorem 3.3, we first reformulate the objective in (10) to a form involving the Stein operator, as stated in Lemma 3.3.

**Lemma 3.3.** (Objective Reformulation) *Denote* $h_k = t_k - t_{k-1}$, *and let* $s_k = \nabla \log f_k(x)$ *denote the score function of* $f_k$. *As* $h_k \to 0$ *and with* $\gamma = \frac{1}{2}$, *the objective in (10) becomes equivalent to the following:*

$$\min_{v_k = v_k(\cdot, 0)} \mathbb{E}_{x \sim f_{k-1}} \left[ -\mathcal{A}_{f_k} v_k + \frac{1}{2} \|v_k\|^2 \right], \quad \mathcal{A}_{f_k} v_k := s_k \cdot v_k + \nabla \cdot v_k.$$

*Proof:*

From the Neural ODE (1) and using Taylor's expansion, we obtain:

$$x(t_k) - x(t_{k-1}) = \int_{t_{k-1}}^{t_k} v_k(x(s), s) ds = h_k v_k(x(t_{k-1}), t_{k-1}) + O(h_k^2) \tag{38}$$

Next, by performing Taylor expansion of $-\log \tilde{f}_k(x(t_k))$ around $t_{k-1}$:

$$\begin{aligned}
-\log \tilde{f}_k(x(t_k)) &= -\log \tilde{f}_k(x(t_{k-1})) + (x(t_k) - x(t_{k-1}))\nabla(-\log \tilde{f}_k(x(t_{k-1}))) + O(h_k^2) \\
&= -\log \tilde{f}_k(x(t_{k-1})) - h_k \nabla \log \tilde{f}_k(x(t_{k-1})) \cdot v_k(x(t_{k-1}), t_{k-1}) + O(h_k^2) \\
&= -\log \tilde{f}_k(x(t_{k-1})) - h_k \, s_k(x(t_{k-1})) \cdot v_k(x(t_{k-1}), t_{k-1}) + O(h_k^2),
\end{aligned} \tag{39}$$

where we define the score function $s_k = \nabla \log f_k = \nabla \log \tilde{f}_k$.

Besides, we also have that:

$$\int_{t_{k-1}}^{t_k} \nabla \cdot v_k(x(s), s) ds = h_k \nabla \cdot v_k(x(t_{k-1}), t_{k-1}) + O(h_k^2). \tag{40}$$

As $h_k \to 0$, we no longer need to divide the time interval, i.e., $S = 1$. The objective function (10) can be then approximated as:

$$\begin{aligned}
&\mathbb{E}_{x \sim f_{k-1}} \left[ -\log \tilde{f}_k(x(t_k)) - \int_{t_{k-1}}^{t_k} \nabla \cdot v_k(x(s), s) \, ds + \frac{1}{2h_k} \|x(t_k) - x(t_{k-1})\|^2 \right] \\
&= \mathbb{E}_{x \sim f_{k-1}} \Bigg[ \left( -\log \tilde{f}_k(x(t_{k-1})) - h_k \, s_k(x(t_{k-1})) \cdot v_k(x(t_{k-1}), t_{k-1}) + O(h_k^2) \right) \\
&\qquad\qquad - \left( h_k \nabla \cdot v_k(x(t_{k-1}), t_{k-1}) + O(h_k^2) \right) + \frac{1}{2h_k} \|h_k v_k(x(t_{k-1})) + O(h_k^2)\|^2 \Bigg] \\
&= \mathbb{E}_{x \sim f_{k-1}} \left[ -\log \tilde{f}_k(x) + h_k \left( -s_k(x) \cdot v_k(x, t_{k-1}) - \nabla \cdot v_k(x, t_{k-1}) + \frac{1}{2} \|v_k(x, t_{k-1})\|^2 \right) + O(h_k^2) \right]
\end{aligned} \tag{41}$$

Since $\mathbb{E}_{x(t_{k-1}) \sim f_{k-1}}[-\log \tilde{f}_k(x(t_{k-1}))]$ is independent of $v_k(x, t)$, as $h_k \to 0$, the minimization of the leading term is equivalent to:

$$\min_{v_k = v_k(\cdot, 0)} \mathbb{E}_{x \sim f_{k-1}} \left[ -\mathcal{A}_{f_k} v_k + \frac{1}{2} \|v_k\|^2 \right], \quad \mathcal{A}_{f_k} v_k := s_k \cdot v_k + \nabla \cdot v_k. \tag{42}$$

Here, $\mathcal{A}_{f_k} v_k$ denotes the Stein operator applied to the vector field $v_k$ under the target density $f_k$.

**Theorem 3.3** (Optimal Velocity Field as Score Difference) *Suppose $h_k \to 0$. Let $f_{k-1}$ and $f_k$ be continuously differentiable on $\mathbb{R}^d$. Assume that $\nabla \cdot v_k(x)$ exists for all $x \in \mathbb{R}^d$, and $\nabla \cdot v_k(x)$, $s_{k-1}$ and $s_k$ belong to $L^2(f_{k-1})$. Assume that the components of $v_k$ are independent and $\lim_{\|x\| \to \infty} f_{k-1}(x)\|v_k(x)\|_2 = 0$. Under these conditions, the minimizer of (10) is:*

$$\lim_{h_k \to 0} v_k^* = s_k - s_{k-1}.$$

*Proof:*

Under the assumptions that $h_k \to 0$ and $\gamma = \frac{1}{2}$, we begin by considering the equivalent minimization objective derived in Lemma 3.3:

$$\min_{v_k} J(v_k) := \min_{v_k} \mathbb{E}_{x \sim f_{k-1}}\left[-\mathcal{A}_{f_k} v_k + \frac{1}{2}\|v_k\|^2\right], \quad \mathcal{A}_{f_k} v_k := s_k \cdot v_k + \nabla \cdot v_k.$$

Expanding the objective functional, we have:

$$\mathbb{E}_{x \sim f_{k-1}}\left[-s_k \cdot v_k - \nabla \cdot v_k + \frac{1}{2}\|v_k\|^2\right] = \int_{\mathbb{R}^d} f_{k-1}(x)\left(-s_k(x) \cdot v_k(x) - \nabla \cdot v_k(x) + \frac{1}{2}\|v_k(x)\|^2\right)dx. \tag{43}$$

Define $B_r = \{x \in \mathbb{R}^d : \|x\| \leq r\}$, and let $\partial B_r$ denote the boundary of $B_r$, which is the sphere of radius $r$. Under the assumption that $\lim_{\|x\| \to \infty} f_{k-1}(x)\|v_k(x)\|_2 = 0$, we have the following:

$$\left|\int_{\mathbb{R}^d} \nabla \cdot (f_{k-1} v_k)\, dx\right| = \lim_{r \to \infty}\left|\int_{B_r} \nabla \cdot (f_{k-1} v_k)\, dx\right| \tag{44}$$

$$= \lim_{r \to \infty}\left|\int_{\partial\{x \in \mathbb{R}^d : \|x\| < r\}} f_{k-1}(x)v_k(x) \cdot n(x)dS(x)\right| \tag{45}$$

$$\leq \lim_{r \to \infty}\int_{\partial\{x \in \mathbb{R}^d : \|x\| < r\}} f_{k-1}\|v_k\|_2\|n(x)\|_2 dS(x) \tag{46}$$

$$= \lim_{r \to \infty}\int_{\partial\{x \in \mathbb{R}^d : \|x\| < r\}} f_{k-1}\|v_k\|_2 dS(x) \tag{47}$$

$$= 0 \tag{48}$$

Therefore, $\int_{\mathbb{R}^d} \nabla \cdot (f_{k-1} v_k)\, dx = 0$. Next, we further expand the divergence theorem:

$$0 = \int_{\mathbb{R}^d} \nabla \cdot (f_{k-1}(x)v_k(x))dx$$

$$= \int_{\mathbb{R}^d} f_{k-1}(x)\nabla \cdot v_k(x)dx + \int_{\mathbb{R}^d} v_k(x) \cdot \nabla f_{k-1}(x)dx \tag{49}$$

$$= \int_{\mathbb{R}^d} f_{k-1}(x)\nabla \cdot v_k(x)dx + \int_{\mathbb{R}^d} v_k(x) \cdot s_{k-1}(x)\, f_{k-1}(x)\, dx$$

Substitute the result back into the objective functional, we have:

$$\mathbb{E}_{x \sim f_{k-1}}\left[-s_k \cdot v_k - \nabla \cdot v_k + \frac{1}{2}\|v_k\|^2\right] = \int_{\mathbb{R}^d} f_{k-1}(x)\left(-s_k(x) \cdot v_k(x) - \nabla \cdot v_k(x) + \frac{1}{2}\|v_k(x)\|^2\right)dx \tag{50}$$

$$= \int_{\mathbb{R}^d} f_{k-1}(x)\left((s_{k-1}(x) - s_k(x)) \cdot v_k(x) + \frac{1}{2}\|v_k(x)\|^2\right)dx. \tag{51}$$

The integrand does not involve $\nabla v_{k,j}(x), j = 1, \cdots d$ and higher-order derivatives. Assuming the components $v_{k,j}, j = 1, \cdots, d$ of $v_k$ are independent, we can take the functional derivative component-wise and set them to zero:

$$\frac{\delta J}{\delta v_k} = f_{k-1}\left(v_k + (s_{k-1} - s_k)\right) = 0, \tag{52}$$

Since $f_{k-1} > 0$ for all $x$, this implies:

$$v_k^* = s_k - s_{k-1}. \tag{53}$$

### A.3. Proofs in Section 5.2

**Density Ratio Estimation (DRE)** *By optimizing the following loss function:*

$$\mathcal{L}_k(\theta_k) = \mathbb{E}_{x(t_{k-1}) \sim f_{k-1}} \left[ \log(1 + e^{-r_k(x_i(t_{k-1}))}) \right] + \mathbb{E}_{x(t_k) \sim f_k} \left[ \log(1 + e^{r_k(x_i(t_k))}) \right],$$

*the model learns an optimal* $r^*(x; \theta_k) = \log \frac{f_{k-1}(x)}{f_k(x)}$.

*Proof:*

Express the loss function as integrals over $x$:

$$\mathcal{L}_k = \int f_{k-1}(x) \log\left(1 + e^{-r_k(x)}\right) dx + \int f_k(x) \log\left(1 + e^{r_k(x)}\right) dx. \tag{54}$$

Compute the functional derivative of $\mathcal{L}_k$ with respect to $r_k$:

$$\frac{\delta \mathcal{L}_k(r_k)}{\delta r_k} = -f_{k-1}(x) \cdot \frac{e^{-r_k(x)}}{1 + e^{-r_k(x)}} + f_k(x) \cdot \frac{e^{r_k(x)}}{1 + e^{r_k(x)}}. \tag{55}$$

Next, we can set the derivative $\delta l_k / \delta r_k(x)$ to zero to find the minimizer $r_k^*(x)$:

$$r_k^*(x) = \ln\left(\frac{f_{k-1}(x)}{f_k(x)}\right). \tag{56}$$

Therefore, by concatenating each $r_k^*(x)$, we obtain

$$r^*(x) = \sum_{k=1}^{K} r_k^*(x) = \log \frac{f_{K-1}(x)}{f_K(x)} \cdot \frac{f_{K-2}(x)}{f_{K-1}(x)} \cdot \ldots \cdot \frac{f_0(x)}{f_1(x)} = \log \frac{f_0(x)}{f_K(x)} = \log \frac{\pi_0(x)}{q^*(x)}, \tag{57}$$

the log density ratio between $\pi_0(x)$ and $q^*(x)$.

## B. Equivalence to Wasserstein gradient flow when $\beta = 1$

In this section, we demonstrate the equivalence between the dynamic optimal transport (OT) objective of AF and the Wasserstein Gradient Flow, under the condition that all $\beta_k$ ($k = 1, 2, \ldots, K$) are set to 1, and a static Wasserstein regularization is used in place of the dynamic Wasserstein regularization introduced in 9.

*Langevin Dynamics and Fokker-Planck Equation:* Langevin Dynamics is represented by the following SDE.

$$dX_t = -\nabla E(X_t)\, dt + \sqrt{2}\, dW_t, \tag{58}$$

where $E(x) = -\log f(x)$ is the energy function of the equilibrium density $f(x, T) = q(x)$. Let $X_0 \sim p_X$ and denote the density of $X_t$ by $\rho(x, t)$. The Langevin Dynamics corresponds to the Fokker-Planck Equation (FPE), which describes the evolution of $\rho(x, t)$ towards the equilibrium $\rho(x, T) = q(x)$, as follows:

$$\partial_t \rho = \nabla \cdot (\rho \nabla E + \nabla \rho), \quad \rho(x, 0) = p_X(x). \tag{59}$$

*JKO Scheme and Wasserstein Gradient Flow:* The Jordan-Kinderlehrer-Otto (JKO) scheme (Jordan et al., 1998) is a time discretization scheme for gradient flows to minimize $\mathrm{KL}(\rho \| q)$ under the Wasserstein-2 metric. Given a target density $q$ and a functional $\mathcal{F}(\rho, q) = \mathrm{KL}(\rho \| q)$, the JKO scheme approximates the continuous gradient flow of $\rho(x, t)$ by solving a sequence of minimization problems. Assume there are $K$ steps with time stamps $0 = t_0, t_1, \cdots, t_K = T$, at each time stamp $t_k$, the scheme updates $\rho_k$ at each time step by minimizing the functional

$$\rho_k = \arg\min_{\rho} \left( \mathcal{F}(\rho, q) + \frac{1}{2\tau} W_2^2(\rho, \rho_{k-1}) \right), \tag{60}$$

where $W_2(\rho, \rho_{k-1})$ denotes the squared 2-Wasserstein distance between the probability measures $\rho$ and $\rho_k$. It was proven in Jordan et al. (1998) that as $h = t_k - t_{k-1}$ approaches 0, the solution $\rho(\cdot, kh)$ provided by the JKO scheme converges to the solution of (59), at each step $k$.

It is straightforward to see that solving for the transport density $\rho_k$ using (60) is equivalent to solving for the transport map $T_k$ via:

$$T_k = \arg\min_{T:\mathbb{R}^d \to \mathbb{R}^d} \left( \mathrm{KL}(T_\# \rho_{k-1} \| q) + \frac{1}{2\tau} \mathbb{E}_{x \sim \rho_{k-1}} \|x - T_k(x)\|^2 \right) \tag{61}$$

Therefore, it is immediately evident that the Wasserstein gradient flow based on the discretized JKO scheme is equivalent to (6) when each $\tilde{f}_k(x)$ is set as the target distribution $q(x)$, i.e., when all $\beta_k$ are set to 1, and the second term in the objective (6) is relaxed to a static $W_2$ regularization instead of a dynamic $W_2$ regularization.

There are some well-established properties regarding the convergence of densities under the Wasserstein Gradient Flow objective when the target distribution $q(x)$ is log-concave. Define $\mathcal{P}_2 = \{P : \int_{\mathbb{R}^d} \|x^2\| dP(x) < \infty\}$ and $\mathcal{P}_2^r = \{P \in \mathcal{P}_2 : P \ll dx\}$.

**Assumption B.1.** For all $n$, the learned velocity field $\hat{v}_n$ guarantees that the mappings $T_n$ is non-degenerate. Additionally, for the time interval $[t_{n-1}, t_n]$, the integrated squared deviation of the velocity field satisfies the inequality

$$\int_{t_{n-1}}^{t_n} \int_{\mathbb{R}^d} \|v - \hat{v}\|^2 \rho \, dx \, dt \leq \epsilon^2, \ \epsilon \in (0, 1).$$

**Assumption B.2.** $F(\rho, q) : \rho \to (-\infty, \infty]$ where $\int_{\mathbb{R}^d} \|x^2\| d\rho(x) < \infty$, is lower semi-continuous; $Dom(F) \subset \mathcal{P}_2^r$; $F(\rho, q)$ is $\lambda$-convex a.g.g. in $\mathcal{P}_2 = \{P : \int_{\mathbb{R}^d} \|x^2\| dP(x) < \infty\}$.

When the energy function $E$ of $q = e^{-E}$ is strongly convex, and $F(\rho, q)$ is chosen as in (61) in our method, Assumption 2 holds true. Under the Assumptions 1-2, the Wasserstein gradient flow converges toward the target distribution at a polynomial rate in terms of the Wasserstein $W_2$ distance. Specifically, the distance between the iterates and the target distribution decays geometrically with each iteration, up to a fixed error term (Cheng et al., 2024).

Furthermore, when additional regularity and integrability conditions are imposed—ensuring boundedness, smoothness, and suitable tail behavior—the convergence behavior improves in a stronger sense. Under these conditions, the gradient flow is shown to converge exponentially fast in the $\chi^2$-divergence measure, meaning that the discrepancy between the iterates and the target decays at an exponential rate (Xu et al., 2024a).

Therefore, when all $\beta_k$ are set to 1 and a static Wasserstein regularization is used in place of the dynamic Wasserstein regularization introduced in (9), our AF retains the well-established convergence properties for log-concave $q$, in terms of both the $W_2$-distance and the $\chi^2$-divergence measure. Furthermore, for non-log-concave densities $q$, we have established in Theorem 3.3 that the difference in the optimal velocity field between two consecutive annealing densities equals the score difference, under the unique dynamic optimal transport (OT) objective of our AF.

## C. Experimental Details

### C.1. Alternative Loss

We empirically observed that replacing the first term $-\log \tilde{f}_k(x(t_k))$ in the objective (10) with the approximation $-h_k \nabla \log \tilde{f}_k(x(t_k)) \cdot v_k$ leads to improved performance. This substitution is motivated by the first-order Taylor expansion of $-\log \tilde{f}_k(x(t_{k-1}))$ around $x(t_k)$, given by:

$$\begin{aligned} -\log \tilde{f}_k(x(t_{k-1})) &= -\log \tilde{f}_k(x(t_k)) - \nabla \log \tilde{f}_k(x(t_k)) \cdot (x(t_{k-1}) - x(t_k)) + \mathcal{O}(h_k^2) \\ &= -\log \tilde{f}_k(x(t_k)) + h_k \nabla \log \tilde{f}_k(x(t_k)) \cdot v_k + \mathcal{O}(h_k^2). \end{aligned} \tag{62}$$

Therefore, the alternative objective to (10) is given by:

$$\min_{v_k(\cdot, t)} \mathbb{E}_{x(t_{k-1}) \sim f_{k-1}} \left[ c_1 - h_k \nabla \log \tilde{f}_k(x(t_k)) \cdot v_k - \int_{t_{k-1}}^{t_k} \nabla \cdot v_k(x(s), s) ds + \alpha \sum_{s=0}^{S-1} \|x(t_{k-1,s+1}) - x(t_{k-1,s})\|^2 \right], \tag{63}$$

where $c_1 = -\log \tilde{f}_k(x(t_{k-1})) + \mathcal{O}(h_k^2)$, noting that $-\log \tilde{f}_k(x(t_{k-1}))$ is independent of $v_k$ given samples $x(t_{k-1}) \sim f_{k-1}$.

We found that this alternative loss led to slightly better performance across several distributions. The alternative objective (63) was used consistently for experiments on the Gaussian Mixture Model, Funnel distribution, and Exponentially Weighted Gaussian. For the Truncated Normal and Bayesian Logistic Regression tasks, we retained the original objective (10) formulation.

### C.2. Hutchinson trace estimator

The objective function in (10) involve the calculation of $\nabla \cdot v_k(x, t)$, i.e., the divergence of the velocity field represented by a neural network. This may be computed by brute force using reverse-mode automatic differentiation, which is much slower and less stable in high dimensions.

We can express $\nabla \cdot v_k(x, t) = \mathbb{E}_{\epsilon \sim p(\epsilon)} \left[ \epsilon^T J_v(x) \epsilon \right]$, where $J_v(x)$ is the Jacobian of $v_k(x, t)$ at $x$. Given a fixed $\epsilon$, we have $J_v(x)\epsilon = \lim_{\sigma \to 0} \frac{v_k(x + \sigma\epsilon) - v_k(x)}{\sigma}$, which is the directional derivative of $v_k$ along the direction $\epsilon$. Therefore, for a sufficiently small $\sigma > 0$, we can propose the following estimator (Hutchinson, 1989; Xu et al., 2024b):

$$\nabla \cdot v_k(x, t) \approx \mathbb{E}_{\epsilon \sim p(\epsilon)} \left[ \epsilon^T \frac{v_k(x + \sigma\epsilon, t) - v_k(x, t)}{\sigma} \right], \tag{64}$$

where $p(\epsilon)$ is a distribution in $\mathbb{R}^{d_Y}$ satisfying $\mathbb{E}[\epsilon] = 0$ and $\text{Cov}(\epsilon) = I$ (e.g., a standard Gaussian). This approximation becomes exact as $\sigma \to 0$. In our experiments, we set $\sigma = 0.02/\sqrt{d}$.

### C.3. Other Annealing Flow settings

*Time steps and numerical integration*

By selecting $K$ values of $\beta$, we divide the original time scale $[0, 1]$ of the Continuous Normalizing Flow (2) and (3) into $K$ intervals: $[t_{k-1}, t_k)$ for $k = 1, 2, \ldots, K$. Notice that the learning of each velocity field $v_k$ depends only on the samples from the $(k-1)$-th block, not on the specific time stamp. Therefore, we can re-scale each block's time interval to $[0, 1]$, knowing that using the time stamps $[(k-1)h, kh]$ yields the same results as using $[0, 1]$ for the neural network $v_k(x, t)$. For example, the neural network will learn $v_k(x, 0) = v_k(x, (k-1)h)$ and $v_k(x, 1) = v_k(x, kh)$, regardless of the time stamps.

Recall that we relaxed the shortest transport map path into a dynamic $W_2$ regularization loss via Lemma 3.2. This requires calculating intermediate points $x(t_{k-1,s})$, where $s = 0, 1, \ldots, S$. We set $S = 3$, evenly spacing the points on $[t_{k-1}, t_k)$, resulting in the path points $x(t_{k-1}), x(t_{k-1} + h_k/3), x(t_{k-1} + 2h_k/3), x(t_k)$. To compute each $x(t_{k-1,s})$, we integrate the velocity field $v_k$ between $t_{k-1}$ and $t_{k-1,s}$, using the Runge-Kutta method for numerical integration. Specifically, to calculate $x(t + h)$ based on $x(t)$ and an intermediate time stamp $t + \frac{h}{2}$:

$$x(t + h) = x(t) + \frac{h}{6}(k_1 + 2k_2 + 2k_3 + k_4),$$

$$k_1 = v(x(t), t), \quad k_2 = v\left(x(t) + \frac{h}{2}k_1, t + \frac{h}{2}\right),$$

$$k_3 = v\left(x(t) + \frac{h}{2}k_2, t + \frac{h}{2}\right), \quad k_4 = v(x(t) + hk_3, t + h)$$

Here, $h$ is the step size, and $v(x, t)$ represents the velocity field.

*The choice of $\beta_k$*

In the experiments on Gaussian Mixture Models (GMMs), we set the number of intermediate $\beta_k$ values to 8, equally spaced such that $\beta_0 = 0, \beta_1 = 1/8, \beta_2 = 2/8, \ldots, \beta_8 = 1$. We chose the easy-to-sample distribution $\pi_0(x)$ as $N(0, I_d)$. Finally, we added 2 refinement blocks. The intermediate distributions are defined as:

$$\tilde{f}_k(x) = \pi_0(x)^{1-\beta_k} \tilde{q}(x)^{\beta_k}.$$

In the experiment on the Truncated Normal Distribution, we did not select $\beta_k$ in the same manner as for the GMM and Exp-Weighted Gaussian distributions. Instead, following the same Annealing philosophy, we construct a gradually transforming bridge from $\pi_0(x)$ to $\tilde{q}(x) = 1_{|x| \geq c} N(0, I_d)$ by setting each intermediate density as:

$$\tilde{f}_k(x) = 1_{\|x\| \geq c/(k+1)} N(0, I_d).$$

The number of intermediate $\beta_k$ values is set to 8.

In the experiment on funnel distributions, we set all $\beta_k = 1$, with the number of time steps set to 8. As discussed in Appendix B, the algorithm becomes equivalent to a Wasserstein gradient descent problem.

In the experiment on 50D Exp-Weighted Gaussian, 20 time steps are used, with 15 intermediate densities and 5 refinement blocks.

*The choice of $\alpha$*

In the experiments on Gaussian Mixture Models (GMMs), funnel distributions, truncated normal, and Bayesian Logistic Regressions, $\alpha$ is uniformly set to $[\frac{8}{3}, \frac{8}{3}, \frac{4}{3}, \frac{4}{3}, \frac{2}{3}, \frac{2}{3}, \frac{2}{3}, \cdots]$. In the experiments on Exp-Weighted Gaussian, $\alpha$ is set to $[\frac{20}{3}, \frac{20}{3}, \frac{20}{3}, \frac{20}{3}, \frac{10}{3}, \frac{10}{3}, \frac{10}{3}, \frac{10}{3}, \frac{5}{3}, \frac{5}{3}, \frac{5}{3}, \frac{5}{3}, 1, 1, 1, 1, 1, 1, 1, 1]$.

*Neural networks and selection of other hyperparameters*

The neural network structure in our experiments is consistently set with hidden layers of size 32-32. During implementation, we observed that when $d \leq 5$, even a neural network with a single hidden layer of size 32 can perform well for sampling. However, for consistency across all experiments, we uniformly set the structure to 32-32.

We sample 100,000 data points from $N(0, I_d)$ for training, with a batch size of 1,000. The Adam optimizer is used with a learning rate of 0.0001, and the maximum number of iterations for each block $v_k$ is set to 1,000.

Different numbers of test samples are used for reporting the experimental results: 5,000 points are sampled and plotted for the experiment on Gaussian Mixture Models, 5,000 points for the experiment on Truncated Normal Distributions, 10,000 points for the experiment on Funnel Distributions, and 10,000 points for the experiment on Exp-Weighted Gaussian with 1,024 modes in 10D space.

## C.4. Bayesian logistic regression

We use a hierarchical Bayesian structure for logistic regression across a range of datasets provided by LIBSVM. The detailed setting of the Bayesian Logistic Regression is as follows.

We adopt the same Bayesian logistic regression setting as described in Liu & Wang (2016), where a hierarchical structure is assigned to the model parameters. The weights $\beta$ follow a Gaussian prior, $p_0(\beta|\alpha) = N(\beta; 0, \alpha^{-1})$, and $\alpha$ follows a Gamma prior, $p_0(\alpha) = Gamma(\alpha; 1, 0.01)$. The datasets used are binary, where $x_i$ has a varying number of features, and $y_i \in \{+1, -1\}$ across different datasets. Sampling is performed from the posterior distribution:

$$p(\beta, \alpha | D) \propto Gamma(\alpha; 1, 0.01) \cdot \prod_{d=1}^{D} N(\beta_d; 0, \alpha^{-1}) \cdot \prod_{i=1}^{n} \frac{1}{1 + \exp(-y_i \beta^T x_i)},$$

We set $\beta_k = 1$ and use 8 blocks to train the Annealing Flow.

During testing, we use all algorithms to sample 1,000 particles of $\beta$ and $\alpha$ jointly, and use $\{\beta^{(i)}\}_{i=1}^{1000}$ to construct 1,000 classifiers. The mean accuracy and standard deviation are then reported in Table 3. Additionally, the average log posterior in Table 3 is reported as:

$$\frac{1}{|D_{\text{test}}|} \sum_{x,y \in D_{\text{test}}} \log \frac{1}{|C|} \sum_{\theta \in C} p(y|x, \theta).$$

## C.5. Importance flow

We report the results of the importance sampler (discussed in Section 5) for estimating $\mathbb{E}_{x \sim N(0,I)} \left[ 1_{\|x\| \geq c} \right]$ with varying $c$ and dimensions, based on our Annealing Flow. To estimate $\mathbb{E}_{x \sim N(0,I)} \left[ 1_{\|x\| \geq c} \right]$, we know that the theoretically optimal proposal distribution which can achieve 0 variance is $\tilde{q}^*(x) = 1_{\|x\| \geq c} N(0, I)$. Then the estimator becomes:

$$\mathbb{E}_{X \sim \pi_0(x)}[h(X)] = \mathbb{E}_{X \sim q^*(x)} \left[ \frac{\pi_0(x)}{q^*(x)} \cdot h(x) \right] \approx \frac{1}{n} \sum_{i=1}^{n} \frac{\pi_0(x_i)}{q^*(x_i)} \cdot h(x_i), \quad x_i \sim q^*(x),$$

where $\pi_0(x) = N(0, I_d)$, $h(x) = 1_{\|x\| \geq c}$ and $q^*(x) = Z \cdot \tilde{q}^*(x)$.

Therefore, the Importance Flow consists of two parts: First, using Annealing Flow to sample from $\tilde{q}^*(x)$; second, constructing a Density Ratio Estimation (DRE) neural network using samples from $\{x_i\}_{i=1}^{n} \sim \tilde{q}^*(x)$ and $\{y_i\}_{i=1}^{n} \sim N(0, I_d)$, as discussed in Section 5.2. The estimator becomes:

$$\frac{1}{n} \sum_{i=1}^{n} DRE(x_i) \cdot h(x_i).$$

The Naive MC results comes from directly using $\{y_i\}_{i=1}^{n} \sim N(0, I_d)$ to construct estimator $\frac{1}{n} \sum_{i=1}^{n} 1_{\|y_i\| \geq c}$. When $c \geq 6$, the Naive MC methods consistently output 0 as the result.

In our experiment, we use a single DRE neural network to construct the density ratio between $\pi_0(x)$ and $q^*(x) = Z \cdot 1_{\|x\| \geq c} N(0, I)$ directly. The neural network structure consists of hidden layers with sizes 64-64-64. The size of the training data is set to 100,000, and the batch size is set to 10,000. We use 30 to 70 epochs for different distributions, depending on the values of $c$ and dimension $d$. The Adam optimizer is used, with a learning rate of 0.0001. The test data size is set to 1,000, and all results are based on 200 estimation rounds, each using 500 samples.

## C.6. Details of other algorithms

The Algorithms 2 and 3 introduce the algorithmic framework of Metropolis-Hastings (MH) and Parallel Tempering (PT) compared in our experiments.

---

**Algorithm 2** Metropolis-Hastings Algorithm

---

1: Initialize $x_0$
2: **for** $t = 1$ to $N$ **do**
3:     Propose $x^* \sim q(x^*|x_{t-1})$
4:     Compute acceptance ratio $\alpha = \min\left(1, \frac{\pi(x^*)q(x_{t-1}|x^*)}{\pi(x_{t-1})q(x^*|x_{t-1})}\right)$
5:     Sample $u \sim \text{Uniform}(0, 1)$
6:     **if** $u < \alpha$ **then**
7:         $x_t = x^*$
8:     **else**
9:         $x_t = x_{t-1}$
10:    **end if**
11: **end for**
12: **return** $\{x_t\}_{t=0}^N$

---

---

**Algorithm 3** Parallel Tempering Algorithm

---

1: Initialize replicas $\{x_1, x_2, \ldots, x_{\text{num\_replicas}}\}$ with Gaussian noise
2: Initialize temperatures $\{T_1, T_2, \ldots, T_{\text{num\_replicas}}\}$
3: **for** $i = 1$ to iterations **do**
4:     **for** $j = 1$ to num_replicas **do**
5:         Propose $x_j^* \sim q(x_j^*|x_j)$ {Using Metropolis-Hastings step for each replica}
6:         Compute acceptance ratio $\alpha_j = \frac{\pi(x_j^*)}{\pi(x_j)}$
7:         Sample $u \sim \text{Uniform}(0, 1)$
8:         **if** $u < \alpha_j$ **then**
9:             $x_j = x_j^*$
10:        **end if**
11:        Store $x_j$ in samples for replica $j$
12:    **end for**
13:    **if** $i \mod \text{exchange\_interval} = 0$ **then**
14:        **for** $j = 1$ to num_replicas $- 1$ **do**
15:            Compute energies $E_j = -\log(\pi(x_j) + \epsilon)$, $E_{j+1} = -\log(\pi(x_{j+1}) + \epsilon)$
16:            Compute $\Delta = \left(\frac{1}{T_j} - \frac{1}{T_{j+1}}\right)(E_{j+1} - E_j)$
17:            Sample $u \sim \text{Uniform}(0, 1)$
18:            **if** $u < \exp(\Delta)$ **then**
19:                Swap $x_j \leftrightarrow x_{j+1}$
20:            **end if**
21:        **end for**
22:    **end if**
23: **end for**
24: **return** samples from all replicas

---

In our experiments, we set the proposal density as $q(x'|x) = \mathcal{N}(x; 0, I_d)$. We use 5 replicas in Parallel Tempering (PT), with a linear temperature progression ranging from $T_1 = 1.0$ to $T_{\max} = 2.0$, and an exchange interval of 100 iterations. For HMC, we set the number of leapfrog steps to 10, with a step size ($\epsilon$) of 0.01, and the mass matrix $M$ is set as the identity matrix. Additionally, we use the default hyperparameters as specified in SVGD (Liu & Wang, 2016), MIED (Li et al., 2023), and AI-Sampler (Egorov et al., 2024). In the actual implementation, we found that the time required for SVGD to converge increases significantly with the number of samples. Therefore, in most experiments, we sample 1000 data points at a time using SVGD, aggregate the samples, and then generate the final plot. The settings for CRAFT, LFIS, and PGPS are discussed alongside each table and figure in Appendix D.

## C.7. Evaluation metrics

To assess the performance of our model, we utilized two key metrics: Maximum Mean Discrepancy (MMD) and Wasserstein Distance, both of which measure the divergence between the true samples and the samples generated by the algorithms.

*Maximum Mean Discrepancy (MMD)*

MMD is a non-parametric metric used to quantify the difference between two distributions based on samples. Given two sets of samples $X_1 \in \mathbb{R}^{n_1 \times d}$ and $X_2 \in \mathbb{R}^{n_2 \times d}$, MMD computes the kernel-based distances between these sets. Specifically, we employed a Gaussian kernel:

$$k(x, y) = \exp\{-\alpha \|x - y\|_2^2\},$$

parameterized by a bandwidth $\alpha$. The MMD is computed as follows:

$$\text{MMD}(X_1, X_2) = \frac{1}{n_1^2} \sum_{i,j} k(X_1^i, X_1^j) + \frac{1}{n_2^2} \sum_{i,j} k(X_2^i, X_2^j) - \frac{2}{n_1 n_2} \sum_{i,j} k(X_1^i, X_2^j),$$

where $k(\cdot, \cdot)$ represents the Gaussian kernel. In our experiments, we set $\alpha = 1/\gamma^2$ and $\gamma = 0.1 \cdot \text{median\_dist}$, where median\_dist denotes the median of the pairwise distances between the two datasets.

*Wasserstein Distance*

In addition to MMD, we used the Wasserstein distance, which measures the cost of transporting mass between distributions. Given two point sets $X \in \mathbb{R}^d$ and $Y \in \mathbb{R}^d$, we compute the pairwise Euclidean distance between the points. The Wasserstein distance is then computed using the optimal transport plan via the linear sum assignment method (from scipy.optimize package):

$$W(X, Y) = \frac{1}{n} \sum_{i=1}^{n} \|X_{r(i)} - Y_{c(i)}\|_2,$$

where $r(i)$ and $c(i)$ are the optimal row and column assignments determined through linear sum assignment.

*Sample-Variance Mean Squared Error across all dimensions*

This metric is reported for Exp-Weighted Gaussian distribution in our experiments. Recall that the distribution of a 50D ExpGauss is:

$$p(x_1, x_2, \cdots, x_{50}) \propto e^{10 \sum_{i=1}^{10} \frac{|x_i|}{\sigma_i^2} + 10 \sum_{i=11}^{50} \frac{x_i}{\sigma_i^2} - \frac{1}{2}\|x\|^2}.$$

To calculate the Mean Squared Error of sample variances across dimensions, we first take the absolute values of the first 10 dimensions, while leaving the values of the remaining dimensions unchanged. The metric is calculated as:

$$\text{MSE} = \frac{1}{D} \sum_{d=1}^{D} (\hat{\sigma}_d^2 - \sigma_d^2)^2, \quad \hat{\sigma}_d^2 = \frac{1}{n-1} \sum_{i=1}^{n} (x_{i,d} - \bar{x}_i)^2.$$

In all experiments, we sample 10,000 points from each model and generate 10,000 true samples from the GMM to calculate and report both MMD and Wasserstein distance. Note that the smaller the two metrics mentioned above, the better the sampling performance.

# D. Additional Experiment Results

We adopt the standard Annealing Flow framework discussed in this paper for experiments on Gaussian Mixture Models (GMM), Truncated Normal distributions, Exp-Weighted Gaussian distributions, and Bayesian Logistic Regressions. For experiments on funnel distributions, we set each $\tilde{f}_k(x)$ as the target $q(x)$, under which the Annealing Flow objective becomes equivalent to the Wasserstein Gradient Flow based on the JKO scheme, as discussed in B. Please refer to C.3 for $\beta_k$ selections.

## D.1. More Results

We begin by presenting the numerical tables for Gaussian Mixture Models (GMMs). In the tables, GMM refers to Gaussian Mixture Models, while wGMM denotes unequally weighted Gaussian Mixture Models. The notation (w)GMM-{number of modes}-{radius of the circle} represents a (w)GMM with the specified number of modes arranged on a circle of the given radius. Note that the Mode-Weight Mean Squared Error (MSE) is also reported in Table 1 (Main Script).

*MMD and Wasserstein Distance*

*Table 4.* MMD results of different methods across the GMM distributions

| | Distributions | **AF** | CRAFT | LFIS | PGPS | PT | SCGD | MIED | AI-Sampler |
|---|---|---|---|---|---|---|---|---|---|
| $d=2$ | GMM-6-8 | $2.38 \times 10^{-3}$ | $\mathbf{2.30 \times 10^{-3}}$ | $1.15 \times 10^{-2}$ | $7.12 \times 10^{-2}$ | $6.27 \times 10^{-2}$ | $9.35 \times 10^{-2}$ | $9.32 \times 10^{-3}$ | $\mathbf{2.34 \times 10^{-3}}$ |
| | GMM-8-10 | $\mathbf{2.45 \times 10^{-3}}$ | $8.98 \times 10^{-2}$ | $2.31 \times 10^{-2}$ | $6.32 \times 10^{-2}$ | $6.48 \times 10^{-2}$ | $1.51 \times 10^{-1}$ | $2.49 \times 10^{-2}$ | $\mathbf{2.61 \times 10^{-3}}$ |
| | GMM-10-12 | $\mathbf{3.01 \times 10^{-3}}$ | $9.06 \times 10^{-2}$ | $8.97 \times 10^{-2}$ | $7.01 \times 10^{-2}$ | $9.01 \times 10^{-2}$ | $1.85 \times 10^{-1}$ | $6.28 \times 10^{-2}$ | $4.02 \times 10^{-3}$ |
| | wGMM-10-12 | $\mathbf{4.95 \times 10^{-3}}$ | $9.96 \times 10^{-2}$ | $1.14 \times 10^{-1}$ | $8.95 \times 10^{-2}$ | $7.02 \times 10^{-2}$ | $2.72 \times 10^{-1}$ | $8.31 \times 10^{-2}$ | $3.19 \times 10^{-3}$ |
| $d=5$ | GMM-6-8 | $\mathbf{5.82 \times 10^{-3}}$ | $9.92 \times 10^{-2}$ | $1.23 \times 10^{-2}$ | $7.81 \times 10^{-2}$ | $8.83 \times 10^{-2}$ | $9.81 \times 10^{-2}$ | $8.01 \times 10^{-3}$ | $7.55 \times 10^{-2}$ |
| | GMM-8-10 | $\mathbf{1.25 \times 10^{-3}}$ | $9.76 \times 10^{-2}$ | $4.52 \times 10^{-2}$ | $7.76 \times 10^{-2}$ | $8.98 \times 10^{-2}$ | $9.63 \times 10^{-2}$ | $3.88 \times 10^{-2}$ | $5.26 \times 10^{-3}$ |
| | GMM-10-12 | $\mathbf{1.57 \times 10^{-3}}$ | $2.14 \times 10^{-1}$ | $7.25 \times 10^{-2}$ | $8.31 \times 10^{-2}$ | $1.18 \times 10^{-1}$ | $1.98 \times 10^{-1}$ | $9.88 \times 10^{-3}$ | $6.37 \times 10^{-3}$ |
| | wGMM-10-12 | $\mathbf{4.31 \times 10^{-3}}$ | $3.95 \times 10^{-1}$ | $8.38 \times 10^{-2}$ | $8.28 \times 10^{-2}$ | $1.05 \times 10^{-1}$ | $1.32 \times 10^{-1}$ | $2.03 \times 10^{-2}$ | $1.87 \times 10^{-2}$ |

*Table 5.* Wasserstein distance results of different methods across the GMM distributions

| | Distributions | **AF** | CRAFT | LFIS | PGPS | PT | SCGD | MIED | AI-Sampler |
|---|---|---|---|---|---|---|---|---|---|
| $d=2$ | GMM-6-8 | $9.38 \times 10^{-1}$ | $\mathbf{9.28 \times 10^{-1}}$ | $8.24 \times 10^{+0}$ | $6.33 \times 10^{+0}$ | $5.71 \times 10^{+0}$ | $9.97 \times 10^{+0}$ | $8.01 \times 10^{-1}$ | $\mathbf{7.92 \times 10^{-1}}$ |
| | GMM-8-10 | $\mathbf{7.22 \times 10^{-1}}$ | $7.57 \times 10^{+0}$ | $8.99 \times 10^{+0}$ | $5.82 \times 10^{+0}$ | $5.98 \times 10^{+0}$ | $1.14 \times 10^{+1}$ | $8.15 \times 10^{-1}$ | $8.95 \times 10^{-1}$ |
| | GMM-10-12 | $\mathbf{8.05 \times 10^{-1}}$ | $8.73 \times 10^{+0}$ | $9.79 \times 10^{+0}$ | $6.09 \times 10^{+0}$ | $7.91 \times 10^{+0}$ | $1.82 \times 10^{+1}$ | $9.35 \times 10^{-1}$ | $8.13 \times 10^{-1}$ |
| | wGMM-10-12 | $\mathbf{9.94 \times 10^{-1}}$ | $9.78 \times 10^{+0}$ | $1.02 \times 10^{+1}$ | $7.88 \times 10^{+0}$ | $6.48 \times 10^{+0}$ | $2.93 \times 10^{+1}$ | $1.06 \times 10^{+0}$ | $8.44 \times 10^{-1}$ |
| $d=5$ | GMM-6-8 | $\mathbf{1.97 \times 10^{+0}}$ | $1.12 \times 10^{+1}$ | $1.01 \times 10^{+1}$ | $7.78 \times 10^{+0}$ | $1.07 \times 10^{+1}$ | $1.13 \times 10^{+1}$ | $2.52 \times 10^{+0}$ | $2.38 \times 10^{+0}$ |
| | GMM-8-10 | $\mathbf{3.33 \times 10^{+0}}$ | $1.98 \times 10^{+1}$ | $3.55 \times 10^{+1}$ | $8.21 \times 10^{+0}$ | $1.53 \times 10^{+1}$ | $2.07 \times 10^{+1}$ | $8.89 \times 10^{+0}$ | $5.53 \times 10^{+0}$ |
| | GMM-10-12 | $\mathbf{2.82 \times 10^{+0}}$ | $2.53 \times 10^{+1}$ | $4.89 \times 10^{+1}$ | $8.75 \times 10^{+0}$ | $1.83 \times 10^{+1}$ | $2.45 \times 10^{+1}$ | $7.89 \times 10^{+0}$ | $3.83 \times 10^{+0}$ |
| | wGMM-10-12 | $\mathbf{3.53 \times 10^{+0}}$ | $3.03 \times 10^{+1}$ | $5.21 \times 10^{+1}$ | $8.64 \times 10^{+0}$ | $2.13 \times 10^{+1}$ | $2.34 \times 10^{+1}$ | $1.13 \times 10^{+1}$ | $9.73 \times 10^{+0}$ |

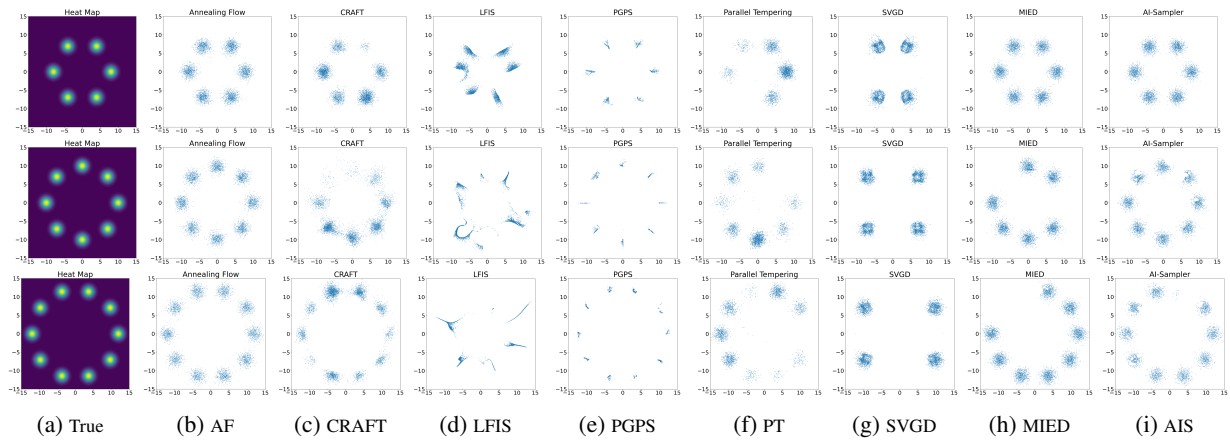

| (a) True | (b) AF | (c) CRAFT | (d) LFIS | (e) PGPS | (f) PT | (g) SVGD | (h) MIED | (i) AIS |

*Figure 4.* Sampling methods for Gaussian Mixture Models (GMM) with 6, 8, and 10 modes arranged on circles with radii $r = 8, 10, 12$. The number of time steps for CRAFT, LFIS, and PGPS is set to 12, the same as for AF.

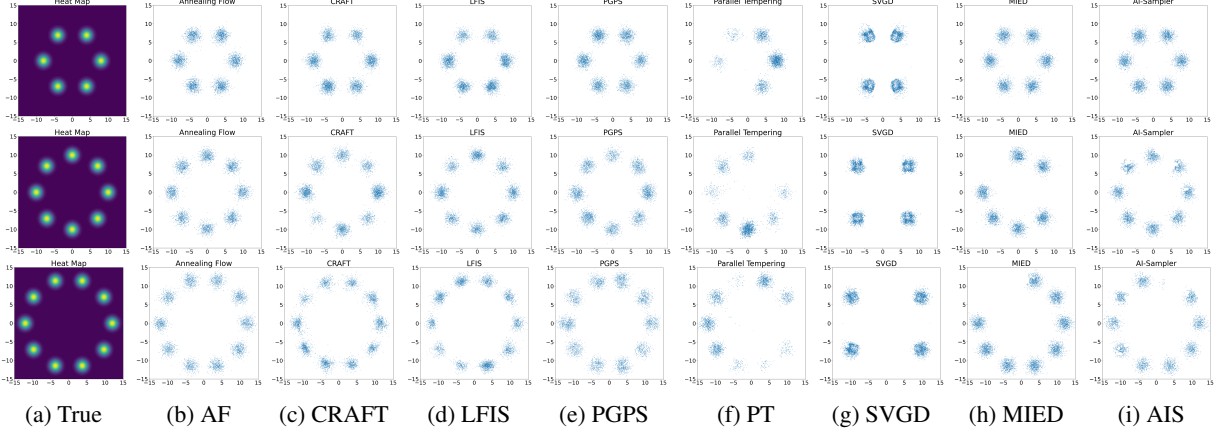

| (a) True | (b) AF | (c) CRAFT | (d) LFIS | (e) PGPS | (f) PT | (g) SVGD | (h) MIED | (i) AIS |

*Figure 5.* Sampling methods for Gaussian Mixture Models (GMM) with 6, 8, and 10 modes arranged on circles with radii $r = 8, 10, 12$. The number of time steps for AF is 12. The number of time steps for CRAFT, LFIS, and PGPS is set as 128, 256, and 128, respectively.

*Gaussian Mixture Models (GMMs)*

We tested each algorithm on GMMs with dimensions ranging from 2 to 5. In the 2D GMM, the modes are arranged in circles with radii $r = 8, 10, 12$. For dimensions higher than 2, the coordinates of the additional dimensions are set to $r/2$.

Figure 5 shows the results when the number of time steps for CRAFT, LFIS, and PGPS is set to 10, the same as for AF. Additionally, Figure 5 presents results where the number of time steps for CRAFT, LFIS, and PGPS is set to 128, 256, and 128, respectively, while the time step for AF remains at 10.

*Truncated Normal Distribution*

Relaxations are applied to the Truncated Normal Distribution in all experiments except for MH, HMC, and PT. Specifically, we relax the indicator function $1_{\|x\| \geq c}$ to $\frac{1}{1+\exp(-k(\|x\|-c))}$. We set $k = 20$ for all experiments. AIS is designed for continuous densities, and we similarly relax the densities in SVGD and MIED, following the approach used in AF. The resulting plots are as follows:

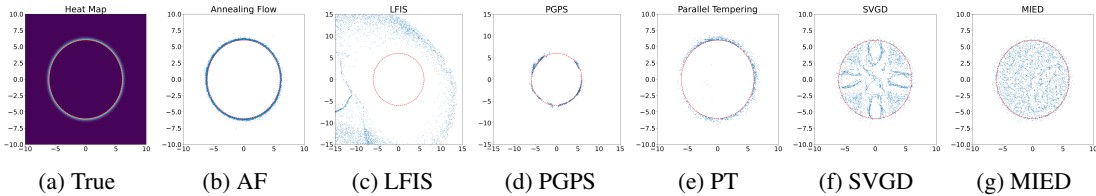

    (a) True      (b) AF      (c) LFIS      (d) PGPS      (e) PT      (f) SVGD      (g) MIED

*Figure 6.* Sampling Methods for Truncated Normal Distributions with Radius $c = 6$, together with the failure cases of SVGD and MIED.

Each algorithm draws 5,000 samples. It can be observed that MCMC-based methods, including HMC and PT, produce many overlapping samples. This occurs because when a new proposal is rejected, the algorithms retain the previous sample, leading to highly correlated sample sets.

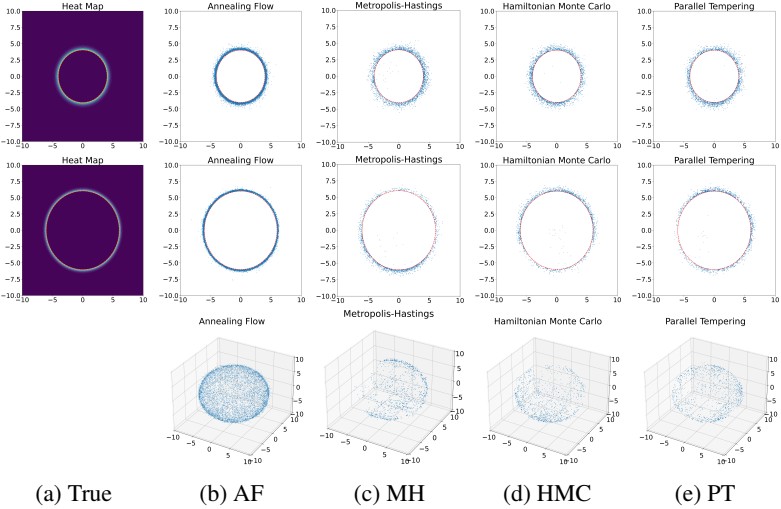

    (a) True      (b) AF      (c) MH      (d) HMC      (e) PT

*Figure 7.* Sampling methods for truncated normal distributions with radii $c = 4, 6$ in 2D space for the first three rows. The last row presents sampling results in *5D* with a radius of 8, projected onto a *3D* space.

*Funnel Distribution*

In the main paper, we present the sampling methods for the funnel distribution with $d = 5$, projected onto a 3D space. To assess the sample quality, here we present the corresponding results projected onto a 2D space, plotted alongside the density heat map.

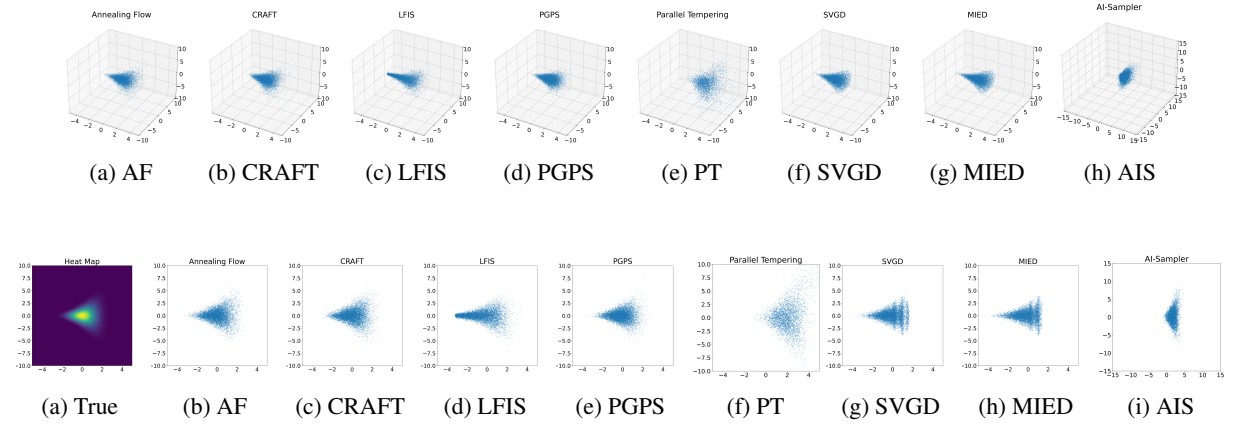

Figure 9. Sampling Methods for Funnel Distribution with $\sigma^2 = 0.81$ in Dimension $d = 5$, projected onto a $d = 3$ Space.

As seen from both figures, our AF method achieves the best sampling performance on the funnel distribution, while other methods, such as MIED and AIS, fail to capture the full spread of the funnel's tail. Additionally, PT, SVGD, and AIS all fail to capture the sharp part of the funnel's shape.

*Exp-Weighted Gaussian Distribution*

Table 2 in the main manuscript presents results for CRAFT, LFIS, and PGPS, using the same number of intermediate time steps as AF. Below, in Table 6, we report the number of modes explored for the 50D Exp-Weighted Gaussian distribution, which has unequal variances across 10 dimensions, when CRAFT, LFIS, and PGPS are trained with 128, 256, and 128 time steps, respectively.

*Table 6.* Number of modes explored in the Exp-Weighted Gaussian distribution by different methods, with CRAFT, LFIS, and PGPS trained with 128, 256, and 128 time steps, respectively. AF is trained with 20 time steps.

|  | True | **AF** | CRAFT | LFIS | PGPS | PT | SVGD | MIED | AIS |
|---|---|---|---|---|---|---|---|---|---|
| $d = 2$ | 4 | **4** | **4** | **4** | **4** | 3.4 | 3.9 | 3.8 | 3.8 |
| $d = 5$ | 32 | **32** | 30.3 | 31.6 | **32** | 25.2 | 28.5 | 28.0 | 28.3 |
| $d = 10$ | 1024 | **1024** | 984.0 | 993.2 | 1002.8 | 233.7 | 957.3 | 923.4 | 301.2 |
| $d = 50$ | 1024 | **1024** | 886.5 | 923.4 | 994.0 | $< 10$ | 916.4 | 890.6 | 125.6 |

We conducted more experiments on asymmetric 50D ExpGaussian distributions with unequal variances across dimensions. In addition to the mode weights' MSE results shown in Table 1 of the main script, we provide an additional metric: the mean squared error of sample variances across dimensions.

Recall that the distribution of 50D ExpGaussian is:

$$p(x_1, x_2, \cdots, x_{50}) \propto e^{10 \sum_{i=1}^{10} \frac{|x_i|}{\sigma_i^2} + 10 \sum_{i=11}^{50} \frac{x_i}{\sigma_i^2} - \frac{1}{2} \|x\|^2}.$$

To calculate the Mean Squared Error of sample variances across dimensions, we first take the absolute values of the first 10 dimensions, while leaving the values of the remaining dimensions unchanged. The metric is calculated as:

$$\text{MSE} = \frac{1}{D} \sum_{d=1}^{D} (\hat{\sigma}_d^2 - \sigma_d^2)^2, \quad \hat{\sigma}_d^2 = \frac{1}{n-1} \sum_{i=1}^{n} (x_{i,d} - \bar{x}_i)^2.$$

Table 7 presents the sample-variance MSE across ExpGauss experiments. The fourth and fifth rows correspond to a 50D ExpGauss where the variances of 2 and 10 randomly selected dimensions are set to $\sigma_i^2 = 0.5$, with the remaining dimensions set to 1. The last two rows correspond to a 50D ExpGauss where the variances of 15 and 20 randomly selected dimensions are uniformly chosen between $\sigma_i^2 = 0.5$ and $\sigma_i^2 = 2$, while the rest are set to 1.

*Importance Flow*

*Table 7.* Sample-Variance Mean Squared Error across dimensions for ExpGauss (1024 modes). The number of time steps for CRAFT, LFIS, and PGPS is set to 20, the same as for AF.

| | Distributions | AF | CRAFT | LFIS | PGPS | SVGD | MIED | AI-Sampler |
|---|---|---|---|---|---|---|---|---|
| $d = 10$ | ExpGauss-1024 | $(\mathbf{1.2} \pm \mathbf{.23}) \times \mathbf{10^{-3}}$ | $(9.8 \pm .49) \times 10^{-3}$ | $(9.8 \pm .75) \times 10^{-3}$ | $(2.5 \pm .37) \times 10^{-2}$ | $(5.2 \pm .50) \times 10^{-2}$ | $(1.7 \pm .48) \times 10^{-2}$ | $(3.9 \pm .75) \times 10^{-2}$ |
| | ExpGauss-1024 | $(\mathbf{1.5} \pm \mathbf{.28}) \times \mathbf{10^{-3}}$ | $(3.7 \pm .71) \times 10^{-2}$ | $(3.5 \pm .56) \times 10^{-2}$ | $(3.0 \pm .67) \times 10^{-2}$ | $(5.7 \pm .49) \times 10^{-2}$ | $(2.5 \pm .63) \times 10^{-2}$ | $(8.2 \pm .76) \times 10^{-2}$ |
| | ExpGaussUV-2-1024 | $(\mathbf{1.6} \pm \mathbf{.26}) \times \mathbf{10^{-3}}$ | $(7.0 \pm .62) \times 10^{-2}$ | $(8.6 \pm .91) \times 10^{-2}$ | $(6.8 \pm .53) \times 10^{-2}$ | $(6.0 \pm .55) \times 10^{-2}$ | $(3.2 \pm .71) \times 10^{-2}$ | $(9.9 \pm .98) \times 10^{-2}$ |
| $d = 50$ | ExpGaussUV-10-1024 | $(\mathbf{2.0} \pm \mathbf{.28}) \times \mathbf{10^{-3}}$ | $(8.8 \pm .80) \times 10^{-2}$ | $(9.2 \pm .98) \times 10^{-2}$ | $(8.9 \pm .73) \times 10^{-2}$ | $(8.1 \pm .90) \times 10^{-2}$ | $(4.8 \pm .82) \times 10^{-2}$ | $(1.6 \pm .25) \times 10^{-1}$ |
| | ExpGaussUV-15-1024 | $(\mathbf{2.1} \pm \mathbf{.31}) \times \mathbf{10^{-3}}$ | $(1.1 \pm .31) \times 10^{-1}$ | $(1.4 \pm .23) \times 10^{-1}$ | $(1.0 \pm .25) \times 10^{-1}$ | $(9.9 \pm .89) \times 10^{-2}$ | $(7.6 \pm .80) \times 10^{-2}$ | $(1.8 \pm .27) \times 10^{-1}$ |
| | ExpGaussUV-20-1024 | $(\mathbf{2.3} \pm \mathbf{.30}) \times \mathbf{10^{-3}}$ | $(2.1 \pm .23) \times 10^{-1}$ | $(2.8 \pm .44) \times 10^{-1}$ | $(1.9 \pm .35) \times 10^{-1}$ | $(1.3 \pm .15) \times 10^{-1}$ | $(9.8 \pm .85) \times 10^{-2}$ | $(2.3 \pm .47) \times 10^{-1}$ |

Table 8 reports the preliminary results of the importance flow (discussed in Section 5) for estimating $\mathbb{E}_{x \sim N(0,I)} \left[ 1_{\|x\| \geq c} \right]$ with varying radii $c$ and dimensions. This estimation uses samples from the experiment on the Truncated Normal Distribution, and thus the results for SVGD, MIED, and AIS cannot be reported. Additionally, we discussed a possible extension of the Importance Flow framework in D.4.

*Table 8.* Comparison of estimation results for $\mathbb{E}_{x \sim N(0,I)}[1_{\|x\|>c}]$ across different radii $c$ and dimensions $d$. Values are reported as mean $\pm$ standard deviation.

| Methods | Radius | $d = 2$ | $d = 3$ | $d = 4$ | $d = 5$ |
|---|---|---|---|---|---|
| True Probability | $c = 4$ | $(3.35) \times 10^{-4}$ | $(1.13) \times 10^{-3}$ | $(3.02) \times 10^{-3}$ | $(6.84) \times 10^{-3}$ |
| | $c = 6$ | $(1.52) \times 10^{-8}$ | $(7.49) \times 10^{-8}$ | $(2.89) \times 10^{-7}$ | $(9.50) \times 10^{-7}$ |
| Importance Flow | $c = 4$ | $(\mathbf{4.04} \pm \mathbf{1.00}) \times \mathbf{10^{-4}}$ | $(\mathbf{1.30} \pm \mathbf{0.23}) \times \mathbf{10^{-3}}$ | $(\mathbf{3.36} \pm \mathbf{0.42}) \times \mathbf{10^{-3}}$ | $(\mathbf{7.86} \pm \mathbf{0.82}) \times \mathbf{10^{-3}}$ |
| | $c = 6$ | $(\mathbf{9.81} \pm \mathbf{4.00}) \times \mathbf{10^{-8}}$ | $(\mathbf{1.51} \pm \mathbf{1.23}) \times \mathbf{10^{-7}}$ | $(\mathbf{2.13} \pm \mathbf{0.87}) \times \mathbf{10^{-7}}$ | $(\mathbf{2.38} \pm \mathbf{3.50}) \times \mathbf{10^{-7}}$ |
| DRE with HMC Samples | $c = 4$ | $(7.56 \pm 4.90) \times 10^{-4}$ | $(2.52 \pm 0.63) \times 10^{-3}$ | $(8.97 \pm 0.91) \times 10^{-3}$ | $(11.2 \pm 1.60) \times 10^{-3}$ |
| | $c = 6$ | $(4.35 \pm 7.21) \times 10^{-7}$ | $(9.01 \pm 2.90) \times 10^{-7}$ | $(1.82 \pm 2.90) \times 10^{-7}$ | $(2.31 \pm 6.21) \times 10^{-7}$ |
| DRE with PT Samples | $c = 4$ | $(6.79 \pm 3.58) \times 10^{-4}$ | $(2.38 \pm 0.54) \times 10^{-3}$ | $(5.78 \pm 7.98) \times 10^{-3}$ | $(9.94 \pm 1.13) \times 10^{-3}$ |
| | $c = 6$ | $(5.37 \pm 9.56) \times 10^{-7}$ | $(8.78 \pm 2.32) \times 10^{-7}$ | $(9.23 \pm 2.51) \times 10^{-7}$ | $(1.98 \pm 7.73) \times 10^{-7}$ |
| Naïve MC | $c = 4$ | $(2.75 \pm 6.00) \times 10^{-4}$ | $(1.18 \pm 1.10) \times 10^{-3}$ | $(2.71 \pm 1.70) \times 10^{-3}$ | $(7.94 \pm 2.60) \times 10^{-3}$ |
| | $c = 6$ | $0$ | $0$ | $0$ | $0$ |

## D.2. Training Efficiency

Table 9 presents the training and sampling times for AF, CRAFT, LFIS, and PGPS in experiments on a 50D Exp-Weighted Gaussian distribution, conducted on a V100 GPU. The training setup includes 100,000 samples, 1,000 training iterations per time step, and a batch size of 1,000 for AF, CRAFT, and PGPS. For LFIS, a batch size of 5,000 is used to ensure good performance. The sampling time is measured for generating 10,000 samples. AF achieves optimal sampling performance with only 20 time steps, compared to other methods requiring up to 256 time steps. Specifically, CRAFT and PGPS were trained with 128 time steps, while LFIS used 256 time steps, ensuring these methods achieved the results shown in Figure 5 and Table 6.

*Table 9.* Total Training and Sampling Time Comparison for the 50D Exp-Weighted Gaussian

| Methods | Training time (mins) | Sampling time (s) |
|---|---|---|
| AF | $\mathbf{14.5} \pm \mathbf{1.3}$ | $\mathbf{2.1} \pm \mathbf{0.5}$ |
| CRAFT | $51.2 \pm 1.8$ | $4.9 \pm 0.6$ |
| LFIS | $86.4 \pm 3.5$ | $6.4 \pm 0.6$ |
| PGPS | $59.7 \pm 2.1$ | $5.2 \pm 0.4$ |

Table 10 reports the training and sampling times per time step (block) for each method. Notably, as AF requires numerical integration over the velocity field, its training time per time step is slightly higher compared to other methods.

*Table 10.* Training and Sampling Time Comparison Per Time Step for the 50D Exp-Weighted Gaussian

| Methods | Training time (mins) | Sampling time (s) |
|---|---|---|
| AF | $0.70 \pm 0.10$ | $0.10 \pm 0.02$ |
| CRAFT | $0.45 \pm 0.09$ | $0.06 \pm 0.01$ |
| LFIS | $0.37 \pm 0.07$ | $0.04 \pm 0.01$ |
| PGPS | $0.44 \pm 0.10$ | $0.04 \pm 0.01$ |

## D.3. Ablation Studies

As reported in the main manuscript, we use 8 intermediate densities and 2 refinement blocks for the GMM experiments, and 15 intermediate densities with 5 refinement blocks for the 50D Exp-Weighted Gaussian experiments. For GMMs, the regularization constant $\alpha$ is set to

$\left[\frac{8}{3}, \frac{8}{3}, \frac{4}{3}, \frac{4}{3}, \frac{2}{3}, \frac{2}{3}, \frac{2}{3}, \cdots\right]$. For Exp-Weighted Gaussian, $\alpha$ is set to $\left[\frac{20}{3}, \frac{20}{3}, \frac{20}{3}, \frac{20}{3}, \frac{10}{3}, \frac{10}{3}, \frac{10}{3}, \frac{10}{3}, \frac{5}{3}, \frac{5}{3}, \frac{5}{3}, \frac{5}{3}, 1, 1, 1, 1, 1, 1, 1, 1\right]$.

In our main experiments, we note that CRAFT, LFIS, and PGPS require 128, 256, and 128 time steps, respectively, to achieve comparable performance. In this section, we conduct additional ablation studies to investigate the role of annealing densities and Wasserstein regularization in ensuring the smoothness and success of Annealing Flow, particularly when fewer time steps are used. We also compare the performance of all NF methods under further reduced time steps.

### D.3.1. SIGNIFICANCE OF ANNEALING DENSITIES AND WASSERSTEIN REGULARIZATION

Here, we conduct experiments on GMMs *without* intermediate densities (i.e., all $f_k(x) = q(x)$) and using 5 blocks. In Figure 10, the second column shows AF with the regularization constant $\alpha$ set to $\left[\frac{4}{3}, \frac{4}{3}, \frac{2}{3}, \frac{2}{3}, \frac{2}{3}\right]$, the third column with $\alpha$ set to $\left[\frac{8}{3}, \frac{8}{3}, \frac{4}{3}, \frac{4}{3}, \frac{4}{3}\right]$, and the fourth column with $\alpha$ set to $\left[\frac{20}{3}, \frac{20}{3}, \frac{8}{3}, \frac{8}{3}, \frac{8}{3}\right]$.

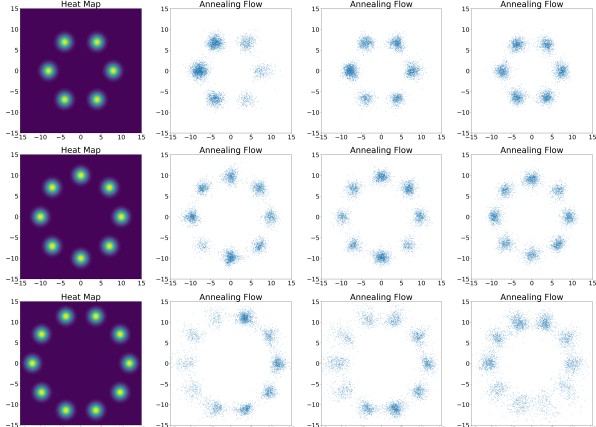

*Figure 10.* Ablation Studies for Gaussian Mixture Models (GMM) with 6, 8, and 10 modes arranged on circles with radii $r = 8, 10, 12$. AF is trained with no intermediate densities and 5 training blocks. The regularization constant $\alpha$ used in the experiments for the three columns is described in the first paragraph of D.3.1.

The figure illustrates an extreme training scenario with no intermediate annealing densities, where the target modes are far separated from the initial density $\pi_0 = N(0, I)$. By comparing this to the successful case, where AF is trained with 8 intermediate densities and 2 refinement blocks (Figure 5), one can immediately see that annealing procedures are essential for successfully handling far-separated modes.

Furthermore, in Figure 10, when no intermediate densities are used, increasing the Wasserstein regularization constant—particularly in the initial blocks—leads to improved results. This experimentally highlights the importance of Wasserstein regularization in our AF objective for ensuring stable performance and significantly reducing the number of intermediate time steps.

*Table 11.* Number of modes explored in the 50D Exponentially-Weighted Gaussian by various methods with different numbers of annealing densities $K$, while keeping the Wasserstein normalizing constant fixed.

|  | AF | CRAFT | LFIS | PGPS |
|---|---|---|---|---|
| $K = 0$ | 18.4 | 4.8 | 3.2 | 6.0 |
| $K = 2$ | 86.7 | 18.0 | 14.2 | 22.8 |
| $K = 4$ | 284.3 | 128.6 | 108.0 | 148.0 |
| $K = 6$ | 808.0 | 256.8 | 186.4 | 424.5 |
| $K = 8$ | 996.2 | 382.0 | 208.4 | 578.8 |
| $K = 10$ | 1024 | 406.2 | 234.0 | 689.0 |

For the challenging 50D Exponentially Weighted Gaussian with 1024 widely separated modes, where the two farthest modes are 63.25 $L_2$ distance apart, annealing procedures are mandatory to ensure success. Table 11 shows the number of modes explored in the 50D Exp-Weighted Gaussian by AF, CRAFT, LFIS, and PGPS as the number of annealing steps $K$ increases. Together with Table 2, it is evident that our AF consistently requires the fewest annealing steps to achieve success in highly challenging scenarios, owing to the $W_2$ regularization of our unique dynamic OT loss.

### D.3.2. PERFORMANCE OF ALGORITHMS WITH EVEN FEWER ANNEALING STEPS

In Figures 3 and 4, the number of time steps for AF, CRAFT, LFIS, and PGPS is set to 10. Here, in Figure 11, we present ablation studies comparing the performance of these four methods when the number of time steps is reduced to 5.

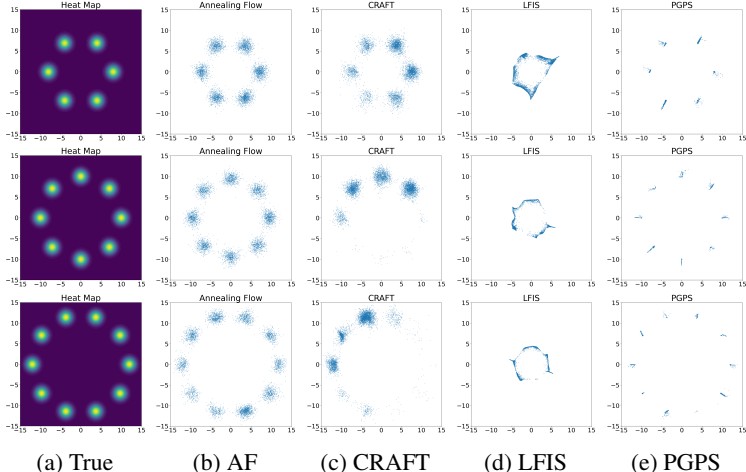

(a) True   (b) AF   (c) CRAFT   (d) LFIS   (e) PGPS

*Figure 11.* Ablation Studies for Gaussian Mixture Models (GMM) with 6, 8, and 10 modes arranged on circles with radii $r = 8, 10, 12$. All methods are trained with 4 intermediate densities and 1 refinement block.

It can be observed that even with half the number of time steps, AF maintains competitive performance on GMMs with 6 and 8 modes and significantly outperforms other methods on GMMs with 10 modes.

*Table 12.* Mode-Weight Mean Squared Error across distributions. The number of time steps for AF, CRAFT, LFIS, and PGPS is reduced to 15 from 20.

|  | Distributions | **AF** | CRAFT | LFIS | PGPS |
|---|---|---|---|---|---|
| $d = 10$ | ExpGauss-1024 | $(\mathbf{9.5 \pm 0.50}) \times \mathbf{10^{-8}}$ | $(4.8 \pm 0.33) \times 10^{-6}$ | $(7.3 \pm 0.74) \times 10^{-6}$ | $(7.8 \pm 0.70) \times 10^{-7}$ |
|  | ExpGauss-1024 | $(\mathbf{1.2 \pm 0.10}) \times \mathbf{10^{-7}}$ | $(7.2 \pm 0.90) \times 10^{-6}$ | $(9.3 \pm 0.82) \times 10^{-6}$ | $(9.8 \pm 0.76) \times 10^{-7}$ |
| $d = 50$ | ExpGaussUV-2-1024 | $(\mathbf{3.3 \pm 0.07}) \times \mathbf{10^{-7}}$ | $(1.7 \pm 0.70) \times 10^{-5}$ | $(2.7 \pm 0.18) \times 10^{-5}$ | $(3.5 \pm 0.49) \times 10^{-5}$ |
|  | ExpGaussUV-10-1024 | $(\mathbf{3.8 \pm 0.11}) \times \mathbf{10^{-7}}$ | $(5.0 \pm 0.82) \times 10^{-5}$ | $(9.8 \pm 0.33) \times 10^{-5}$ | $(7.6 \pm 0.73) \times 10^{-5}$ |

We conducted similar ablation studies on the 50D Exp-Weighted Gaussian with 1024 widely separated modes, reducing the number of time steps from 20 to 15. Table 12 presents the mode-weight MSE in Exp-weighted Gaussian across different dimensions and unequal variances, when the number of time steps is reduced to 15 from 20 for all NF methods. AF still successfully captures all 1024 modes, with slightly higher Mode-Weight MSEs compared to the values reported in Table 1. In contrast, other methods, including CRAFT, LFIS, and PGPS, perform much worse than AF.

## D.4. Possible Extensions of Importance Flow

The importance flow discussed and experimented with in this paper requires a given form of $\pi_0(x)$, and thus, a given form of $\tilde{q}^*(x) = \pi_0(x) \cdot |h(x)|$ for estimating $\mathbb{E}_{X \sim \pi_0(x)}[h(X)]$. In our experimental settings, $\tilde{q}^*(x) = 1_{\|x\| \geq c} N(0, I_d)$ can be regarded as the Least-Favorable-Distribution (LFD). We conducted a parametric experiment for the case where $\tilde{q}^*(x)$ has the given analytical form.

However, we believe future research may extend this approach to a distribution-free model. That is, given a dataset without prior knowledge of its distribution, one could attempt to learn an importance flow for sampling from its Least-Favorable Distribution (LFD) while minimizing the variance. For example, in the case of sampling from the LFD and obtaining a low-variance IS estimator for $P_{x \sim \pi(x)}(\|x\| \geq c)$, one may use the following distribution-free loss for learning the flow:

$$\min_\theta \frac{1}{n} \sum_{i=1}^n \left[ 1\{T(x_i; \theta) \leq c\} \cdot \|T(x_i; \theta) - c\|^2 \right] + \gamma \int_0^1 \|v(x(t), t; \theta)\|^2, \tag{65}$$

where the first term of the loss pushes the dataset $\{x_i\}_{i=1}^n$ towards the Least-Favorable tail region, while the second term ensures a smooth

and cost-optimal transport map. Note that the above loss assumes no prior knowledge of the dataset distribution $\pi(x)$ or the target density $q(x)$.

Xu et al. (2024c) has also explored this to some extent by designing a distributionally robust optimization problem to learn a flow model that pushes samples toward the LFD $Q^*$, which is unknown and learned by the model through a risk function $\mathcal{R}(Q^*, \phi)$. Such framework has significant applications in adversarial attacks, robust hypothesis testing, and differential privacy. Additionally, the recent paper by Ribera Borrell et al. (2024) introduces a dynamic control loss for training a neural network to approximate the importance sampling control. We believe that by designing an optimal control loss in line with the approaches of these two papers, one can develop a distribution-free Importance Flow for sampling from the LFD of a dataset while minimizing the variance of the adversarial loss, which can generate a greater impact on the fields of adversarial attacks and differential privacy.

