# OpenReview forum: "Annealing Flow Generative Models Towards Sampling High-Dimensional and Multi-Modal Distributions"
_ICML.cc/2025/Conference — ICML 2025 poster_

### Official Review · Reviewer_mLHv · 2025-03-07

**Overall Recommendation:** 3

**Summary:**

This paper proposes Annealing Flow, a method based on continuous normalizing flow to sample from high dimensional multi-modal distribution. The authors provide a efficient training and sampling algorithm, which can be also applied to Monte-Carlo estimation. Various experiments are conducted to verify the efficiency of AF compared to previous sampling methods.

**Claims And Evidence:**

Yes

**Essential References Not Discussed:**

[1] Qiu, Yixuan, and Xiao Wang. "Efficient multimodal sampling via tempered distribution flow." Journal of the American Statistical Association 119.546 (2024): 1446-1460.

[2] Maurais, Aimee, and Youssef Marzouk. "Sampling in unit time with kernel fisher-rao flow." arXiv preprint arXiv:2401.03892 (2024).

see weakness part for details

**Experimental Designs Or Analyses:**

Yes, in Section 6

**Methods And Evaluation Criteria:**

Yes

**Other Comments Or Suggestions:**

see weakness part

**Other Strengths And Weaknesses:**

## Strengths

The paper is written clearly and all the mathematic derivations are easy to follow. Apart from developing algorithms, the authors also present some theoretical insights of annealing flow, in the limit of infinitesimal time. The proposed method is useful to handle multi-modal distribution.

## Weaknesses
### 1.
The idea is not very novel overall, and the paper lacks discussions and comparisons with some important previous works. For example, [1] considers similar annealing flow, utilizing L2 distance to train the map between $f_{k-1}$ and $f_k$ and claiming that KL divergence used in this paper will suffer from mode collapse. [2] uses kernel trick to estimate the map instead of neural network, which may save the training cost to some extent. The authors should include more comparisons, either from theoretical side or empirical side.

### 2.
The numerical experiments are not very convincing due to lack of more high-dimensional challenging distributions. The highest dimension in the experiments is merely 50, which is still too low in modern applications of machine learning. I suggest the authors include more high-dimensional experiments such as Bayesian neural networks.

### 3.
The theoretical claims in Proposition 3.3 and 3.4 are not mathematically rigorous. For example, eq (12) holds only in the sense of limit when $h\to 0$ and thus it's not mathematically sound to directly write eq (12). A better way to express this is, e.g., $\lim_{h\to 0} v_k^*-(s_k-s_{k-1}=0$.

[1] Qiu, Yixuan, and Xiao Wang. "Efficient multimodal sampling via tempered distribution flow." Journal of the American Statistical Association 119.546 (2024): 1446-1460.

[2] Maurais, Aimee, and Youssef Marzouk. "Sampling in unit time with kernel fisher-rao flow." arXiv preprint arXiv:2401.03892 (2024).

**Questions For Authors:**

Are the neural networks trained in block-wise manner also stored sperately? If the annealing timesteps are high and the size of neural network is large, the storage cost is not negligible.

**Relation To Broader Scientific Literature:**

It is generally related to machine learning community

**Theoretical Claims:**

Yes. Some claims are not mathematically rigorous, see weakness part

---

> ### Author Rebuttal · Authors · 2025-03-31
>
> We sincerely thank the reviewer for your valuable comments and suggestions! Please see below for our response to your concerns.
>
> > Summary of Review: The paper lacks novelty and misses key related works. [1] proposes an annealed NF using L2 distance. [2] employs kernel methods instead of neural networks to reduce training costs. The authors should provide more thorough comparisons.
>
> Thank you for bringing these works to our attention! The novelty of our work lies in the 1. computational efficiency, 2. stability, and 3. unique theoretical property enabled by our annealing dynamic OT objective, along with 4. strong sampling performance—most notably on the challenging 50D ExpGauss with 1024 far-separated modes, well beyond existing sampling densities baselines.
>
> While we tried best to survey a broad range of NFs/CNFs, and compared three very recent NF methods (plus four beyond NF/CNF scope), we acknowledge that some relevant works have been missed.
>
> We've now implemented **[1] ("TemFlow")** and evaluated it on GMMs (with radius 8, 10, 12) and the extreme 50D ExpGauss distribution. Below is a link to the figures and tables:
>
> **Anonymous link**: https://drive.google.com/file/d/1M_o4XF5440n0G_k_PaHioncx-ro1mHhd/view?usp=sharing.
>
> We sincerely appreciate your time in reviewing the link—*it took considerable effort and we would be grateful for your feedback!*
>
> Summary on [1]:
>
> - *Architecture sensitivity*:
>
> TemFlow is an NF method whose performance depends on its spline layer design (e.g., number/type of bins, bounds), which must be adapted for each target density. In contrast, AF consistently uses a fixed 32-32 MLP without any network-specific tuning. As shown in link~Fig. 3, suboptimal spline design can significantly degrade TempFlow’s performance.
>
> - *Annealing steps*:
>
> TemFlow’s official implementation uses 100 annealing on GMMs (radius=4), while our AF achieves comparable performance using 10 steps on a harder GMM (radius=12). Fig. 1 compares TempFlow when matched to our step count.
>
> - *Efficiency*:
>
> TemFlow trains each annealing block quickly (0.39 min vs. our 0.70 min), but needs more steps—its total training time on 50D ExpGauss reaches 49.6 min versus 14.5 min for AF.
>
>
> Additionally, [2] is conceptually similar to SVGD and MIED, and we compared the latter two in our paper. [2] is a kernel-based, gradient-driven method and not directly related to NF/CNF. For their comparison with SVGD, please see the first paragraph of their Page 5. We’ll provide further context as we investigate further.
>
> We also evaluated **NUTS** mentioned by Reviewer Eiy4. We'd be grateful if you could also review that response!
>
> > The experiments are not very convincing due to lack of more high-dimensional challenging distributions [...].
>
> Thanks for the thoughtful question! Our primary contribution lies in advancing statistical sampling methods, but not targeting specific real-world datasets. To our knowledge, prior work has not attempted sampling from the highly challenging distributions such as 50D ExpGauss distribution:
> $$p(x_1,x_2,\cdots,x_{50}) \propto e^{10\sum_{i=1}^{10}\frac{|x_{i}|}{\sigma_{i}^{2}}+10\sum_{i=11}^{50}\frac{x_{i}}{\sigma_{i}^2}-\frac{1}{2}\|x\|^{2}},$$
> which has 1024 far-separated modes (nearest two separated by 20, farthest by 63.25).
>
> For context, prior works primarily evaluate on lower-dimensional or less multimodal distributions: **TempFlow** [1] experiments on a 2D GMM with 8 close modes, a Copula distribution with 4 modes, and samples from up to a 32D latent space (usually unimodal). **FisherRao** [2] explores up to 20D funnel and 2D close modes. **LFIS** (ICML 2024, compared in our work) focuses on a 2D GMM with 9 close modes and a 10D funnel. **AI-Sampler** (ICML 2024, compared in our work) tests on a 2D circular GMM (radius 5) and Bayesian regression up to 21 dimensions.
>
> Statistical sampling is fundamental in areas like statistical physics, rare event analysis, and Bayesian modeling. Beyond testing on challenging distributions, we introduce the novel *Importance Flow framework* for rare-event probability estimation (Table 9); and through *Hierarchical Bayesian Modeling* on real datasets ranging from 8 to 61 dimensions (Table 3).
>
> > Other Comments:
>
> We have replaced the Eq. (12) in Prop. 3.4 with the following:
>
> $$\lim_{h_k \to 0} v_{k}^*=s_k-s_{k-1}.$$
>
> Thank you for the detailed observation!
>
> > **Summary of our Contributions** (summarized in more detail at the end of our response to **Reviewer aCqJ**):
>
> 1. Our dynamic OT objective significantly reduces the number of intermediate annealing steps, compared to most recent NF/CNFs works.
> 2. Prop. 3.4 establishes $\lim_{h_k \to 0} v_{k}^*=s_k-s_{k-1}$, a desired property unique to our AF.
> 3. Our experiments on highly challenging densities show AF's superior performance towards multi-modal and high-dimensional sampling - on tasks that go beyond prior works.
>
> Thank you once again for your valuable time and thoughtful feedback—we truly appreciate it!

---

> > ### Comment · Reviewer_mLHv · 2025-04-03
> >
> > Thanks for the authors' feedback. The comparison results between AF and TempFlow are convincing, but it seems that the authors didn't answer the question "Are the neural networks trained in block-wise manner also stored sperately?" I'm willing to raise my score accordingly.

---

> > > ### Author Response · Authors · 2025-04-03
> > >
> > > Dear Reviewer mLHv,
> > >
> > > Thank you sincerely for your reply to our response! We apologize for not fully addressing your concerns earlier due to character limitations.
> > >
> > > > "Are the neural networks trained in block-wise manner also stored sparately?"
> > >
> > > Yes, in our algorithm, each $v_k$ is trained sequentially after $v_1, v_2, \dots, v_{k-1}$ have been trained. Consequently, each $v_k$ network needs to be stored separately. However, enabled by our unique dynamic OT loss, the total number of annealing steps ($v_k$) is significantly reduced compared to other recent NF/CNFs works. Additionally, our algorithm requires much simpler neural network architecture —a simple NN with two 32-32 hidden layers.
> > >
> > > In our challenging 50D ExpGauss experiments, there are 20 $v_k$ trained in total, leading to a total storage of:
> > >
> > > $$20 \ \text{NNs} \times 4338 \  \text{parameters/NN} \times 4 \ \text{bytes/param} = 347,040 \ \text{bytes} \approx 339 \text{KB},$$
> > >
> > > which, including optimizer states and other training components, gives approximately **1.9-2.2 MB** overall storage size in total for our **AF**.
> > >
> > > This highlights another key advantage of our dynamic OT-based loss: fewer annealing steps and simpler neural network structures compared to other normalizing flow methods. For instance, **other CNF methods** compared in our work require the storage capacities **up to 120 MB**, due to the needs for a 128-128 MLP and much more annealing steps.
> > >
> > > *The stored CNF parameters also provide substantial advantages for sampling tasks*. With these parameters available, sampling is reduced to numerical computations, rapidly generating 10,000 samples in just 2.1 seconds for the 50-dimensional case (Tables 10 and 11). This stands in clear contrast to MCMC methods, which typically require long mixing times and frequently struggle in multimodal scenarios.
> > >
> > > We sincerely appreciate the time and effort you’ve dedicated to reviewing our work!

---

### Official Review · Reviewer_Eiy4 · 2025-03-13

**Overall Recommendation:** 3

**Summary:**

This paper proposed a new flow-based sampler from continuous target density functions via combining several ideas.
More specifically, they introduce a method that they call Annealing Flow (AF) by using continuous normalizing flows trained with dynamic optimal transport objective function. They use Wasserstein regularization and their method is guided by annealing the target pdf.

**Claims And Evidence:**

To some extent.
My criticism is as follows:
While in the title and throughout the paper they claim their method is suitable for high-dimensional targets, their experimental models are quite toyish and low-dimensional. Clearly, reporting results for high-dimensional models would support their claim better (or would show its limitations that would still be valuable).
Also, (apart from one visual experiment in the supplementary) they do not compare their method against MCMC-based methods. For instance, it would be interesting to see the mode switches and MSE versus the model dimension for the AF and NUTS sampler.  Also, it would be good to report the AF's training time and compare MCMC versus AF (and other flow-based methods) if the total time budget is limited and fixed.

**Essential References Not Discussed:**

Not that I am aware of

**Experimental Designs Or Analyses:**

Yes but see my  comments on "Claims And Evidence"

**Methods And Evaluation Criteria:**

Yes but see my  comments on "Claims And Evidence"

**Other Comments Or Suggestions:**

N/A

**Other Strengths And Weaknesses:**

As I mentioned earlier, my main criticism is not testing the performance of their method on high-dimensional models (e.g. d > 100) as well as not comparing it with MCMC.
Clearly, every algorithm has its limitations, but it would be good to provide experimental results that give an idea of cases where AF outperforms MCMC (say NUTS) and cases where the opposite is true.

**Questions For Authors:**

N/A

**Relation To Broader Scientific Literature:**

Combining several state-of-the-art algorithms in an smart way and proposing a more robust flow-based method that (apparently) can be used in high-dimensional and multi-model models.

**Theoretical Claims:**

I did not closely check the correctness of the proofs. Also, it is hard for me to convince myself that the proposed density ratio estimation (Section 5.2) will perform better than the formula in line 289. It would be valuable if they would compare the performance of these two approaches.

---

> ### Author Rebuttal · Authors · 2025-03-31
>
> Thank you very much for your helpful review and thoughtful comments! We address your concerns point by point below:
>
> > It is hard for me to convince myself that the proposed density ratio estimation (Section 5.2) will perform better than the formula in line 289.
>
> Thank you for your detailed observation on the extended Importance Flow framework. The traditional normalized Importance Sampling (IS) estimator in line 289 is generally biased. For example, in our IS experiment on estimating $\mathbb{E}_{X \sim \pi_0}[h(X)]$ with $h(x) = \text{1}(||x||>c)$, $\pi_0 (x) = N(x;0,I)$, the resulting optimal proposal $\tilde{q}^* (x) =  \text{1}(||x||>c) \cdot N(x;0,I)$. We can immediately see the self-normalized IS estimator is:
>
> $$\hat{I}_{N}= \sum \frac{\pi_0 (x_i)}{\tilde{q}^* (x_i)} h (x_i) / \sum \frac{\pi_0 (x_i)}{\tilde{q}^* (x_i)} = \sum 1 / \sum \frac{1}{\text{1}(||x||>c)} =  \sum 1 / \sum 1 = 1, \text{where} \ x_i \sim q^* (x).$$
>
> The estimator always equals to 1 in this case. However, the true probability we want to estimate is $P_{X \sim N(0,I)} (||X||>c)$.
>
> However, our Density Ratio Estimation (DRE) network introduced in Section 5.2 can theoretically recover the exact ratio $r^* (x) = \log \frac{\pi_0 (x)}{q^* (x)}$, where $q^* (x)$ is the normalized target density. This allows us to directly construct an importance sampling (IS) estimator:
> $$\hat{I} = \sum \frac{\pi_0 (x_i)}{q^* (x_i)} h (x_i)=\sum \exp (r^* (x_i))\cdot h(x_i).$$
> As shown in the preliminary results in Table 9, the Importance Flow yields a rare-event probability estimate that closely matches the true value, rather than defaulting to 1 as in $\hat{I}_{N}$.
>
>
>
> > My main criticism is not testing the performance of their method on high-dimensional models (e.g. d > 100) as well as not comparing it with MCMC. It would be good to provide experimental results that give an idea of cases where AF outperforms MCMC (say NUTS) and cases where the opposite is true.
>
> Thank you very much for your thoughtful feedback! We included comparisons with two MCMC-based methods in our experiments: **Parallel Tempering (PT)**, a classical tempered MCMC, and **AI-Sampler**, a neural-network-assisted MCMC. A functional comparison with MCMC is also briefly discussed in Section 4.2. That said, we agree that further comparison is valuable and have added more detailed analysis now in our draft. To summarize:
>
> - *Sampling Efficiency*:
>
> Unlike MCMC, which does not require pre-training but often suffers from long mixing times, AF requires one-time offline training, after which sampling becomes a deterministic, efficient numerical computations. We reported **training and sampling time** of AF in Tab. 10 and Tab. 11.
> - *Performance in High Dimensions*:
>
> In our 50D ExpGauss experiment with 1024 far-separated modes, PT captures <10 modes and AIS captures around 100 (Table 3), illustrating MCMC’s limitations in exploring highly multimodal, high-dimensional spaces. AF significantly outperforms in this setting.
> - *Sample Balance*:
>
> Even when MCMC reaches multiple modes, it often fails to maintain proper sample weighting due to the stochastic nature of mode-hopping.
>
> Further, we have now implemented **NUTS**, and below is a link of figures and tables:
>
> **Anonymous Link:** https://drive.google.com/file/d/1vyciBojsTKuRfMfw0GZ6LYfMXx49g7Fe/view?usp=sharing
>
> *It took us great efforts so we sincerely appreciate your time in reviewing it!*
>
> As shown in link~Tab. 1 and 2, MCMC methods—NUTS, PT, and AIS—fail immediately on the 50D target with 1024 well-separated modes. This is due to the exponential growth of the effective search space (roughly $O(\exp (d))$) for a MC, making exploration of all far-separated modes infeasible in high dimensions.
>
> > For your main criticism on not testing the performance on high-dimensional models (e.g. d > 100):
>
> Notably, the 50D ExpGauss (nearest two separated by 20, farthest by 63.25) we study is substantially more challenging than prior benchmarks.
>
> For context, among very recent methods: **TemFlow** (referred by Reviewer mLHv) experiments on a 2D GMM with 8 close modes, a Copula distribution with 4 close modes, and samples from up to a 32D latent space (normally uni-modal). **Fisher-rao** (referred by Reviewer mLHv) explores up to 20D funnel and 2D closely aligned modes. **LFIS** focuses on a 2D GMM with 9 close modes and a 10D funnel. **AI-Sampler** tests on a 2D circular GMM (radius 5) and Bayesian logistic regression up to 21 dimensions.
>
> In summary, we compared the three major sampling paradigms—MCMC (NUTS, AI-Sampler, PT), particle-based methods (SVGD, MIED), and normalizing flows (CRAFT, LFIS, PGPS, TempFlow)—on significantly more challenging densities than prior works.
>
> We have also summarized our key contributions at the end of our response to **Reviewer aCqJ**, and we’d be grateful if you could take a look!
>
> Thank you sincerely for your thoughtful feedback—it helped us clarify both our MCMC comparisons and overall contributions!

---

### Official Review · Reviewer_zGCn · 2025-03-14

**Overall Recommendation:** 3

**Summary:**

The paper proposes a new method to learn a vector field v(x,t) such that the neural ODE dx = v(x,t) dt approximates the optimal transport between two given distributions p and q. This is laudable, because scalable and accurate optimal transport solvers in high dimensions are still an important research topic. The paper derives a tractable loss that leads to an efficient learning algorithm. The new method appears to be superior to existing algorithms on several benchmarks.

Unfortunately, the presentation contains severe errors and shortcomings that might not be fixable in a rebuttal phase, see below.

EDIT: In light of the rebuttal discussion, I raised my score.

**Claims And Evidence:**

The paper defines the "annealing flow" as the sequence of functions f_k(x) = p(x)^{1-beta_k} q(x)^{beta_k} with two predefined distributions p and q and a sequence of inverse temperatures beta_0 = 0 < beta_1 < ... < beta_K = 1 (equation (4)). However, the claim that the f_k(x) are distributions is false, because the RHS is not normalized for 0 < beta_k < 1 (this is easily seen by setting p = normal(0,1) and q = normal(1,1)). Consequently, the KL divergence in equation (6) is undefined, as it requires normalized distributions.

This mistake may not have severe consequences, because the authors subsequently allow q to be unnormalized, which requires the introduction of an explicit normalization constant. Since this constant is independent of the model, it has no influence on the gradient of the loss in equation (8) -- an incorrect normalization constant would thus be inconsequential. However, this is not quite clear, because the definition of tilde(E)_k after equation (7) makes no sense: The constant tilde(Z)_k and the energy function E_k(x) are both undefined at this point. The proof of proposition 3.1 in the appendix does not resolve the riddle: Contrary to equation (4), the f_k in the proof are defined by the probability path induced by the continuity equation (2). The thus defined f_k are normalized, but require that the vector field v(x,t) is already known. In addition, the proof uses a normalization constant Z_k instead of tilde(Z)_k, which is much more plausible.

These contradictions must be fixed.

**Essential References Not Discussed:**

See above.

**Experimental Designs Or Analyses:**

See above.

**Methods And Evaluation Criteria:**

The new method is compared to seven alternatives, using challenging synthetic data distributions, and wins most of the time. However, it seems that only one of these experiments uses an existing benchmark protocol, so results are hard to compare with the literature.

In the qualitative experiment on a 2D Gaussian mixture (figure 3), all methods except the proposed one perform terribly. This is highly implausible, as similar experiments with satisfactory results are frequently reported in the literature. It looks as if the hyperparameters of the competition have not been tuned properly -- this would be unacceptable. To be fair, it could also be a consequence of forcing all methods to use the same network architecture for comparability, but even if this were the case, the choice of network architecture would be suspicious. Of course, this also sheds a bad light on the other experimental comparisons.

**Other Comments Or Suggestions:**

See above.

**Other Strengths And Weaknesses:**

In additional to the errors discussed above, I'd like to raise the following issues:
* The authors hierarchically discretize time first into segments [t_{k-1}, t_k] and then into subsegments t_{k-1,s}. However, it is never discussed why this setup is necessary or beneficial or superior to alternatives, and what the underlying trade-offs are.

* In order to drop the constant c from equation (8), the objective in equation (10) must be an argmin. Moreover, it would be nice to illustrate how the divergence term and the energy term counterbalance each other such that this objective has a fixed point at the optimum (instead of converging to a trivial solution).

* In section 4.1, the authors implement the Hutchinson estimator in terms of discrete directional derivatives. However, the underlying expression epsilon^T J_v epsilon (see appendix C.1) can be evaluated exactly using Jacobian-vector primitives from the autodiff library. The proposed implementation introduces an unnecessary additional approximation error.

* Section 5 on importance flows seems out of scope, given that there are only preliminary results which are only reported in the appendix. Leave this topic for future work!

**Questions For Authors:**

See above.

**Relation To Broader Scientific Literature:**

The paper reviews a considerable body of existing literature. However, it is surprising that "Optimal flow matching" (Kornilov et al. arXiv:2403.13117) is not mentioned and not compared against.

The claim in the introduction "Discrete normalizing flows often suffer from mode collapse" is highly questionable. This only happens when either the hyperparameters are not chosen properly, or the network is trained with reverse KL. Properly chosen NF architectures trained with the forward KL (the standard approach) do not exhibit mode collapse.

**Theoretical Claims:**

See above.

---

> ### Author Rebuttal · Authors · 2025-03-31
>
> Thank you for your thoughtful review and careful attention to our mathematical developments! We acknowledge that some symbols lacked rigor and appreciate you pointing them out! Please see below for our response to your concerns on both math rigor and experimental comparisons.
>
> > The paper defines $f_k(x) = \pi_{0}(x)^{1-\beta_{k}} q(x)^{\beta_k}$ in Eq. (4). [...] However, f_k(x) are not normalized distributions. Consequently, the KL divergence in Eq. (6) is undefined, as it requires normalized distributions.
>
> Thanks for the careful review! You are absolutely right—our original definition of $f_{k}(x)$ in (4) was unnormalized. We have now slightly revised $f_k(x)$ in (4) by:
>  $$\tilde{f}_k(x)=\pi_0 (x)^{1-\beta_k} \tilde{q} (x)^{\beta_k},$$
>
> $$f_k (x)= Z_k \tilde{f}_k (x),$$
> where $\tilde{q} (x)$ is the unnormalized target density and $Z_k = \frac{1}{\int \tilde{f}_k (x)dx}$ is an unknown constant.
>
> - **The only change that needs to be made is in how we define $Z_k$.** Our algorithm solely relies on $\tilde{f}_k$, which is defined the same as original. The constant $Z_k$ of the normalized $f_k$ appears only in the constant term $c$ in Eq. (8) of Prop. 3.1, given by:
>
> $$c = E_{x\sim \rho(x(t_{k-1}),t_{k-1})}[\log \rho(x(t_{k-1}),t_{k-1})]- \log Z_k,$$
> which is a constant independent of our algorithm.
>
> - With the revised definition of $f_k$, the KL divergence in Eq. (6) is well-defined, and all subsequent propositions and algorithmic steps remain correct.
>
> Thanks for the detailed observation!
>
> > Following $f_k (x)$, the $E_k $, $\tilde{E}_k$, and $\tilde{Z}_k $ are unclear [...]. Contrary to Eq. (4), the f_k in the proof are defined by the probability path (2). And it requires the vector field $v(x,t)$ already known.
>
> Based on $f_k (x)$ above, the energy $E_k (x)$ is still defined as $E_k (x) = -\log f_k (x)$, and we've now made this clear in the draft. We have also removed the redundant definitions of $\tilde{E}_k$ and $\tilde{Z}_k $, now consistently used $Z_k$ in Prop. 3.1, and replaced all occurrences of $\tilde{E}_k$ in the draft with:
> $\tilde{E}_k \mapsto -\log \tilde{f}_k (x).$
>
> The density $\rho(\cdot, t)$ evolves according to Eq. (2). Under the constraints of the dynamic OT (3), we have: $\rho(\cdot, t_{k-1}) = f_{k-1},\ \rho(\cdot, t_k) = f_k$. We adopt an efficient block-wise training (as opposed to end-to-end), in which each $v_k (x,t)$ is trained sequentially after $v_1,\cdots, v_{k-1}$ have been learned, as given in Alg. 1.
>
> > Your concerns about Methods And Evaluation Criteria:
>
> We would like to clarify: **Fig. 3** shows GMM results for CRAFT, LFIS, and PGPS using the **same number of annealing steps (8–10)** as our AF, ensuring a fair comparison on **computational efficiency**. In contrast, **Fig. 5** presents these same methods with their **official settings**—using significantly more steps (128 for CRAFT and PGPS, 256 for LFIS) with PGPS using Langevin adjustments. They perform well in Fig. 5, but at a higher computational cost, as detailed in Tables 10 and 11.
>
> Our experiments focus exclusively on *well-separated modes*, where methods without annealing - PT, SVGD, MIED, and AIS—naturally struggle. In contrast, *prior works commonly evaluate on closely aligned GMM modes*: e.g., radius 1 in LFIS (their paper's Fig. 2), radius 4 in both AI-Sampler (their Fig. 3) and the TemFlow (their Fig. 3) referenced by Reviewer mLHv.
>
> By comparison, in our GMM settings (Fig. 3 and 5), modes are separated by radius 12. And in the extremely challenging 50D ExpGauss, by distances ranging from 20 to 63.25.
>
> We also did ablation studies on **AF without annealing** (Fig. 10 and Tab. 12), where our performance drops notably. This shows the necessity of Annealing.
>
> To fairly evaluate AF’s *efficiency and performance*, results reported in **Tab. 3** (main text), **Tab. 7**, and **Fig. 5** (appendix) for other NFs uses their official (and much higher) annealing steps to reflect their full performance.
>
> The purpose of **Fig. 3** is only to *compare performance under matched annealing steps to highlight AF's efficiency*. We've now added clearer references to Fig. 5 in the main text to make this distinction.
>
> > Response to other concerns:
>
> 1. "Optimal Flow Matching" is designed for image generation and requires target samples for training, not target densities as in statistical sampling. We've further implemented **TemFlow** mentioned by **Reviewer mLHv** and **NUTS** mentioned by **Reviewer Eiy4**. Please see the link in the responses!
> 2. $t_{k-1,s}$ is defined for $W_2$ discretization purpose in Prop. 3.2. We've made it clearer and more concise.
> 3. Our code implemented both torchdiff and Hutchinson for the divergence. torchdiff requires more computation with little improvement.
>
> > Finally:
>
> We have summarized our key contributions at the end of our response to Reviewer aCqJ, and we’d be grateful if you could take a look!
>
> Thank you sincerely for your detailed review—it has indeed helped make our paper stronger!

---

> > ### Comment · Reviewer_zGCn · 2025-04-02
> >
> > > In contrast, Fig. 5 presents these same methods with their official settings—using significantly more steps (128 for CRAFT and PGPS, 256 for LFIS) with PGPS using Langevin adjustments. They perform well in Fig. 5, but at a higher computational cost, as detailed in Tables 10 and 11.
> >
> > You might want to move (part of) figure 5 to the main text, space permitting (e.g. by eliminating Section 5). It greatly contributes to the trustworthiness of your method. (In addition, make clear that figures 4 and 5 are in the Appendix.) You should also say explicitly in the caption that 128 steps for CRAFT etc. are the settings recommended by the original authors.
> >
> > > "Optimal Flow Matching" is designed for image generation
> >
> > This doesn't do justice to the paper. E.g. their figure 6 corresponds to your figure 1, and they explicitly establish an algorithm that requires only one step to generate the data.
> > The same example also features in figure 2 of arXiv:1808.04730, another one-step model that generates the distribution very accurately.
> > Please put your method into perspective properly. The two cited methods clearly do not "naturally struggle" with this setting.
> >
> > > The authors hierarchically discretize time first into segments [t_{k-1}, t_k] and then into subsegments t_{k-1,s}. However, it is never discussed why this setup is necessary or beneficial or superior to alternatives, and what the underlying trade-offs are.
> >
> > Your answer to this question is not yet satisfactory. It is clear that the discretization is needed in Prop. 3.2. My question concerns why your particular choices in this regard are good and outperform other possibilities.

---

> > > ### Author Response · Authors · 2025-04-02
> > >
> > > Dear Reviewer zGCn,
> > >
> > > We sincerely appreciate your in-time reply to our rebuttal!
> > >
> > > > You might want to move (part of) figure 5 to the main text, space permitting (e.g. by eliminating Section 5). It greatly contributes to the trustworthiness of your method.
> > >
> > > Yes, thank you very much for the thoughtful suggestion! We plan to move Fig. 5 to the main text, potentially replacing Fig. 3 and moving Fig. 3 to the appendix.
> > >
> > > We also appreciate your suggestion regarding Section 5 (Importance Flow). This section aims to provide the community with a potential framework for constructing an unbiased importance sampling estimator using flows. This is a relatively unexplored direction and offers an additional metric to assess sample quality. That said, we will consider removing it to better comply with the page limit!
> > >
> > >
> > > > Concerns about "Optimal Flow Matching":
> > >
> > > Thanks for the clarification. We should not have said that "Optimal Flow Matching" is solely designed for image generation, as it also includes illustrative GMM experiments:
> > >
> > > However, a **key distinction** is: methods "Optimal Flow Matching" and arXiv:1808.04730 **train the flow model using samples from the target distribution**, and they are not designed for statistical sampling. For example, to learn a flow that maps to a GMM (as in their Fig. 2), they have access to samples from that GMM during training. As a result, *such methods cannot be used to sample challenging densities like the 50D ExpGauss in our experiments*, where obtaining target samples for training is infeasible.
> > >
> > > In contrast, our Annealing Flow **(AF) requires only the target density, not target samples**. During training, AF samples exclusively from the initial distribution $\pi_0(x)= N(x; 0,I)$ and learns the velocity field $v_k$ using only forward-evolved particles $x(t_{k-1})$, but never relying on target samples $x(t_k)$.
> > >
> > > Our method is specifically designed for the **statistical sampling** setting, where the target density is known but sampling from it directly for training is infeasible.
> > >
> > > Besides, we have removed the words “Discrete normalizing flows often suffer from mode collapse”. Thank you for pointing out this misleading sentence!
> > >
> > > > Response to concerns about the discretization in Prop. 3.2:
> > >
> > > We apologize for not addressing it more clearly earlier due to character limits.
> > >
> > > As you noted, discretization is necessary in Proposition 3.2 because the integral $\int_{t_{k-1}}^{t_k} \|v_k\|^2 \, dt$ cannot be directly computed in closed form, and approximating it numerically would require significantly finer discretization.
> > >
> > > One could approximate the transport cost using only the endpoints, e.g., $\|x(t_k) - x(t_{k-1})\|^2$. However, incorporating intermediate points $x_{t_{k-1},s}$ allows for enforcing a smooth and more optimal transport path and reflects the *dynamic* nature of the transport cost, rather than reducing it to a static movement between two fixed points.
> > >
> > > In our experiments, we consistently use two intermediate points for regularization: $x(t_{k-1} + h_k/3)$ and $x(t_{k-1} + 2h_k/3)$. This approach already yields strong empirical performance without increasing the computational burden much.
> > >
> > > We hope we have correctly understood and addressed your questions! If you have any further concerns, please edit your reply to rebuttal above, and we would be more than happy to answer more!
> > >
> > > If you feel your questions have been addressed, we would be grateful if you consider updating your review. Thank you sincerely for your time spent in our work!

---

### Official Review · Reviewer_aCqJ · 2025-03-17

**Overall Recommendation:** 4

**Summary:**

The authors devise a new technique for sampling from high-dimensional multi-modal distributions. The assumed setting is that we are given an unnormalized analytical form of the density we wish to sample from. The proposed technique, dubbed Annealing Flow, is based on a continuous normalizing flow guided by an annealing procedure and is trained with a dynamic optimal transport objective. This has several benefits such as enabling stable training without a score-matching objective and cutting down the number of annealing steps needed to achieve state-of-the-art performance on the tested datasets.

## Update after rebuttal:
I find the authors answers satisfying and appreciate the extra effort put in to improve this work clarity and benchmarking. Therefore, I'm happy to change my score to "Accept" to increase their chances for acceptance.

**Claims And Evidence:**

Yes, the experiments do seem to prove empirically that methods/claims are valid in comparison to the baselines. Nonetheless, as I'm unfamiliar with the tasks in this field, I'm not certain how representative these unnormalized densities/sampling tasks are.

**Essential References Not Discussed:**

I'm not very familiar with this line of work and do not follow it closely. However, it does seem the authors did their due diligence and cited the existing literature properly.

**Experimental Designs Or Analyses:**

I haven't thoroughly checked the details as I'm unfamiliar with the examined densities, but the overall experiments section seems sound.

**Methods And Evaluation Criteria:**

I'm not very familiar with this line of work, but after skimming through some of the references in the paper, I noticed that the authors performed extensive comparisons with quite a few representative recent baselines.

**Other Comments Or Suggestions:**

General suggestion: I think overall the method goes through so many objectives until it arrives at the final one, and this part could be made significantly more concise and clearer for the reader. For instance, a small paragraph explaining the task at the beginning of the method section would be very helpful.
Caught preceding shorthand: In line 26 in the abstract the MC shorthand is used before being defined as Monte Carlo.

**Other Strengths And Weaknesses:**

In terms of mathematical rigor, the paper is written with a very high standard. Nonetheless, the writing could be made clearer and more concise for the benefit of the average reader, and some parts could be relayed to the appendix and replaced with more intuitive explanations. For example, how do you calculate $\tilde{E}_{k}(x(t_k))$ is still unclear to me in the current version.

**Questions For Authors:**

1) How do you calculate $\tilde{E}_{k}(x(t_k))$?
2) You mentioned your method requires more expensive pertaining, is this quantified somewhere?

**Relation To Broader Scientific Literature:**

The paper aims to provide an easier-to-train and overall more efficient sampler for multi-modal high-dimensional distributions. The paper seems theoretically solid as far as I can tell, however, I'm not sure how novel and impactful are the contributions regarding the dynamic optimal transport objective, and if these had been proposed elsewhere in prior works (other than the referenced ones).

**Theoretical Claims:**

I did go over some of the proofs in Appendix A and as far as I can tell they seem sound. Also, the theoretical results regarding the optimal velocity field with infinitesimal annealing steps make sense and feel overall intuitive.

---

> ### Author Rebuttal · Authors · 2025-03-31
>
> Thank you sincerely for your time and thoughtful review! Below, we respond to each of your questions in detail. We also provide a summary of our key contributions and the representativeness of our comparison experiments at the end of this response.
>
> > Nonetheless, the writing could be made clearer and more concise for the benefit of the average reader, and some parts could be relayed to the appendix and replaced with more intuitive explanations.
>
> Thanks for the helpful comments! We agree that the algorithmic development and theoretical claims can be made more concise and accessible. Following your suggestions, we have made the following revisions:
>
> 1. *Reorganized Fig. 2*:  We moved Fig. 2 earlier in the paper to accompany the introduction of $f_k$ and $\tilde{f}_k$, making the exposition more illustrative.
>
> 2. *Clarified definitions*: We corrected the definition of $Z_k$ associated with $f_k$, as pointed out by **Reviewer zGCn**; We added Wasserstein distance $W_2$ definition in Section 2 to help general readers follow the paper more easily.
>
> 3. *Removed redundancy*: We eliminated the redundant definition of $\tilde{E}_k$ and replaced it with $-\log \tilde{f}_k$; We removed the redundant definition of $\tilde{Z}_k$;
>
> 4. *Streamlined the objective*: We made the final objective (Eq. 10) more explicit and moved Proposition 3.3 (Objective Reformulation) into Proposition 3.4 to clarify that the reformulation is used solely for the derivation of Proposition 3.4, avoiding unnecessary repetition.
>
>
> > How do you calculate $\tilde{E}_k (x(t_k))$?
>
> We originally defined $\tilde{E}_k (x) = -\log \tilde{f}_k$ under (7), but upon revision, found this definition unnecessary and have replaced it with the direct expression $-\log \tilde{f}_k$ throughout the paper. Given the unnormalized target density $\tilde{q}(x)$ and the definition of $\tilde{f}_k$ in Eq. (4), the form $\tilde{E}_k (x(t_k))=-\log \tilde{f}_k$ is known.
>
> In objective (10), the expectation $\mathbb{E}$ is taken over sample $x(t_{k-1})$ from the density $f_{k-1}$, while the evaluation of $\tilde{E}_k (x(t_k))=-\log \tilde{f}_k (x(t_k))$ requires the sample at the later time step $x(t_k)$. Therefore, a numerical integration is needed to compute $x(t_k)$, as shown in Eq. (13):
>
> $$x(t_k) = x(t_{k-1}) + \int_{t_{k-1}}^{t_k} v_k (x,s) ds,$$
> where $v_k$ is the learnable velocity field optimized in the objective (10).
>
> We referenced the computation of $x(t_{k-1})$ in Line 5 of Alg. 1. We have now clarified this step further in the revised paper.
>
> > You mentioned your method requires more expensive pertaining, is this quantified somewhere?
>
> Yes, we reported training and sampling times for AF and other Normalizing Flows (NFs) in Tables 10 and 11. While AF requires one-time pre-training, it can be done offline and reused for efficient sampling—producing 10,000 samples in 50D space in just 2.1 seconds (given in Tab. 10). In contrast, MCMC methods require significantly longer mixing times and yield poor performance.
>
> We have also included AF vs. MCMC comparisons in our response to **Reviewer Eiy4**. We would greatly appreciate it if you could take a look. We hope these have address your concerns!
>
> Besides, We have now further implemented **NUTS** (mentioned by Reviewer Eiy4) and **TemFlow** (mentioned by Reviewer mLHv), and would sincerely appreciate it if you could take a look at the anonymous links provided in our responses to them!
>
> > Below, we summarize our **key contributions** and further highlight the **depth of our experimental evaluations**:
>
> 1. *Algorithmic novelty*:
>
> Our annealing-based dynamic Optimal Transport (OT) objective is novel in the current literature, combining the strengths of both annealing and dynamic OT to yield several key advantages outlined below.
>
> 2. *Greatly improved training efficiency and sampling performance*:
>
> With our dynamic OT loss, AF achieves superior performance using only 10 annealing steps—compared to 256 in LFIS, 128 in CRAFT and PGPS, and 100 in TemFlow. As shown in Tables 10 and 11, this results in significantly faster training.
>
> 3. *Theoretical advantages*:
>
> In Proposition 3.4, we establish that the optimal velocity field satisfies $\lim_{h \to 0} v_{k}^* = s_k - s_{k-1}$, i.e., the score difference between successive annealing densities. This is a desirable property uniquely enabled by our annealing-based dynamic OT objective.
>
> 4. *Experimental evaluation*:
>
> We benchmark AF against leading methods across three sampling mainstreams: MCMC (PT, AI-Sampler, and the new NUTS mentioned by Reviewer Eiy4), gradient-based methods (SVGD, MIED), and NF-based methods (CRAFT, LFIS, PGPS, and the new TempFlow noted by Reviewer mLHv). Notably, we also take a big step towards sampling very challenging cases, the 50D ExpGauss distribution with 1024 far-separated modes (separated by distances ranging from 20 to 63.25).
>
>
> We sincerely appreciate your valuable time and feedback! We look forward to any further comments you may have!

---

### Decision · Program_Chairs · 2025-05-01

**Decision:**

Accept (poster)

**Comment:**

This paper proposes Annealing Flow, a normalizing flow method for sampling from high-dimensional and multi-modal distributions. It builds upon continuous normalizing flows (CNFs) by introducing an annealing-guided dynamic optimal transport (OT) objective. It improves training efficiency and stability compared to previous NF methods. Empirical results demonstrate strong performance across a diverse set of baselines, particularly on high-dimensional benchmarks.

Despite some concerns regarding clarity and novelty, most reviewers recognized the method’s technical depth and consistent empirical improvements.